# X-ray-activated polymerization expanding the frontiers of deep-tissue hydrogel formation

Hailei Zhang [1] ✉, Boyan Tang[1], Bo Zhang[1], Kai Huang[2], Shanshan Li[1], Yuangong Zhang[1], Haisong Zhang[3], Libin Bai[1], Yonggang Wu [1], Yongqiang Cheng[1], Yanmin Yang[4] ✉ & Gang Han [2] ✉

Photo-crosslinking polymerization stands as a fundamental pillar in the domains of chemistry, biology, and medicine. Yet, prevailing strategies heavily rely on ultraviolet/visible (UV/Vis) light to elicit in situ crosslinking. The inherent perils associated with UV radiation, namely the potential for DNA damage, coupled with the limited depth of tissue penetration exhibited by UV/Vis light, severely restrict the scope of photo-crosslinking within living organisms. Although near-infrared light has been explored as an external excitation source, enabling partial mitigation of these constraints, its penetration depth remains insufficient, particularly within bone tissues. In this study, we introduce an approach employing X-ray activation for deep-tissue hydrogel formation, surpassing all previous boundaries. Our approach harnesses a low-dose X-ray-activated persistent luminescent phosphor, triggering on demand in situ photo-crosslinking reactions and enabling the formation of hydrogels in male rats. A breakthrough of our method lies in its capability to penetrate deep even within thick bovine bone, demonstrating unmatched potential for bone penetration. By extending the reach of hydrogel formation within such formidable depths, our study represents an advancement in the field. This application of X-ray-activated polymerization enables precise and safe deep-tissue photo-crosslinking hydrogel formation, with profound implications for a multitude of disciplines.

Photo-crosslinking has emerged as a powerful technique for fabricating three-dimensional polymer networks with broad applications across diverse fields, including biomedicine[1–4]. It offers precise spatial and temporal control over hydrogel synthesis, making it invaluable for wound healing[5,6], drug delivery[7], tissue repair[8–10], bioengineering[11], and bioimaging[12]. The hallmark advantage lies in the ability to trigger polymerization on-demand using light, enabling highly controlled synthesis[13,14]. However, the current approach relies on ultraviolet/ visible (UV/Vis) light (200 ‑ 780 nm) for activation, which poses significant limitations[15–18]. UV/Vis light exhibits restricted tissue penetration, typically reaching only a few millimeters[19], thus severely hampering its utility in biological systems. Moreover, concerns surrounding DNA damage associated with UV light raise critical safety considerations, particularly for patient applications[20]. Recent attempts have explored the use of near-infrared (NIR) light with longer wavelengths to enhance penetration depth to some extent[21–23]. There are

[1]College of Chemistry & Materials Science, Hebei University, Baoding 071002, P. R. China. [2]Department of Biochemistry and Molecular Pharmacology, University of Massachusetts Medical School, Worcester, Massachusetts, MA 01605, USA. [3]Affiliated Hospital of Hebei University, Baoding 071000, P. R. China. [4]College of Physics Science and Technology, Institute of Life Science and Green Development, Hebei Key Lab of Optic-electronic Information and Materials, Hebei University, Baoding 071002, P. R. China. ✉e-mail: zhanghailei@hbu.edu.cn; mihuyym@163.com; Gang.Han@umassmed.edu

several other mechanisms to confer injectability upon hydrogels and facilitate their gelation within deep tissues, largely relying on temperature-, pH, magnet-, shearing-, or chemical-based stimuli. These stimuli, while well suited for use in some target applications, usually suffer from a range of disadvantages that render them inappropriate for use in many in vivo applications. The use of injectable temperature-sensitive hydrogels usually suffers from unexpected gelation inside the needles warmed by the body temperature, especially for the injection in deep tissues[24]. Magnetic stimuli can interfere with some clinical therapeutics and medical imaging technologies[25]. pH stimuli are restricted in the regions that match the material's pH-responsive range[26]. Chemical stimuli require contact with the material and are difficult to spatially control[25]. Shear-thinning hydrogels can be injected under the shear stress during injection and rapidly return to their elastic behavior after removal of shear[27], while the gelation process is difficult to control after being injected into the body.

Nonetheless, existing crosslinking methods fall short in achieving deep-tissue penetration with spatial and temporal control, especially within bone structures. Consequently, the repair and regeneration of bone tissue have remained formidable challenges within the realm of photo-crosslinking hydrogel systems. Therefore, the need for other approaches that address these limitations and enable deep-tissue hydrogel formation is imperative for advancing biomedical research and therapeutic interventions. Compared to light, X-rays, a form of electromagnetic waves extensively utilized in biomedical applications, exhibit remarkable penetration capabilities through both bones and human tissue (Fig. 1)[28–31]. Moreover, clinical X-ray radiation offers precise control over exposure time and location[32,33], crucial factors for enabling deep tissue cross-linking reactions necessary for biomedical hydrogel formation.

In this study, we present a breakthrough X-ray-controlled polymer crosslinking method, referred to as Xcrosslinking, that achieves hydrogel formation within deep tissues and bone structures with advantages including highly spatial controllability, noncontact external stimuli, and uncritical demand for the environment. To accomplish this, we develop a biocompatible X-ray-activated visible persistent luminescent emitting phosphors (X-PLNPs) serving as an in situ energy transducer[34–36], initiating photo-crosslinking reactions at deep tissue sites, including within bones. Specifically, we utilize halloysite nanotubes (HNTs), natural tubular nanomaterials with good water dispersibility[37,38], charge on the external surface[39], low toxicity[40], and ease of modification[41], as templates for the synthesis of HNTs-based X-PLNPs (HNTs@YF$_3$:Tb$^{3+}$) through a facile hydrothermal method. These Tb$^{3+}$-doped X-PLNPs exhibited well-controlled size, desirable water-dispersity, intense persistent luminescence, excellent biocompatibility, and robust stability. We demonstrate that HNTs@YF$_3$:Tb$^{3+}$ effectively store energy upon low-dose X-ray excitation, releasing visible luminescence as a persistent light source. This emitted visible light from HNTs@YF$_3$:Tb$^{3+}$ serves as a trigger for on-demand, in situ photo-crosslinking reactions, enabling hydrogel formation. We validate the performance of our system through in vitro experiments and animal models. Our results demonstrate high biocompatibility and the potential safety of this approach, as it enables in situ crosslinking activation in deep tissue. This groundbreaking strategy holds tremendous promise for various biomedical applications requiring deep tissue hydrogel formation, offering a frontier in the field of X-ray-activated polymer crosslinking.

## Results and discussion
### Harnessing persistent luminescent emitting phosphors for enhanced functionality
In order to leverage the unique properties of X-PLNPs, we strategically immobilized the chelate agent ethylenediaminetetraacetic disodium salt (EDTA-2Na) onto the surface of HNTs, facilitating the formation of X-PLNP complexes (Fig. 2a). This modification involved the

treatment of aminated HNTs (HNTs-NH$_2$) with excess EDTA-2Na, resulting in the covalent attachment of the ethylenediaminetetraacetic acid (EDTA) moiety to the nanotube through a conventional esterification reaction. Comprehensive characterization using solid-state nuclear magnetic resonance (NMR, Supplementary Fig. 3a and b), Fourier transform infrared spectroscopy (FTIR, Supplementary Fig. 4), and thermogravimetric analysis (TGA, Supplementary Fig. 5) confirmed the successful generation of EDTA-derivatized HNTs (HNTs-EDTA) while preserving the fundamental structure and composition of the nanotubes.

To fabricate HNTs@YF$_3$:Tb$^{3+}$ nanocomposites (Fig. 2a), we employed a hydrothermal reaction conducted at elevated temperature and pressure. Crucially, a slightly acidic environment was maintained throughout the process to ensure successful nanocomposite formation. The combination of NH$_4$F and Y(NO$_3$)$_3$·6H$_2$O served as precursors for the synthesis of YF$_3$, the host material, while the rare-earth dopant Tb$^{3+}$ acted as the activator. The resulting HNTs@YF$_3$:Tb$^{3+}$ exhibited excellent water dispersibility (Fig. 2b).

To confirm the chemical composition of HNTs@YF$_3$:Tb$^{3+}$, we performed X-ray photoelectron spectroscopy (XPS) analysis. A comparison between the XPS spectra of HNTs and HNTs@YF$_3$:Tb$^{3+}$ revealed discernible changes in accordance with the formation of the nanocomposite (Fig. 2c and Supplementary Table 1). After modification and hydrothermal treatment, the Si/Al ratio of HNTs@YF$_3$:Tb$^{3+}$ was determined to be 1.42, indicating successful modification via the silane coupling agent. The XPS analysis of HNTs@YF$_3$:Tb$^{3+}$ exhibited new peaks corresponding to the presence of fluorine (F1s, Fig. 2d) and yttrium (Y 3d, Fig. 2e). Notably, the XPS spectrum also revealed the presence of an N 1s peak at a binding energy of 397.4 eV, originating from the EDTA moiety, and a Y 3s peak at 402.6 eV (Fig. 2f). Additionally, changes in the C 1s region, corresponding to C−C bonds (285.5 eV) and C=O bonds (289.5 eV), were observed in the XPS spectrum of HNTs@YF$_3$:Tb$^{3+}$ (Fig. 2g). The existence of terbium in HNTs@YF$_3$:Tb$^{3+}$ was confirmed by the doublet peak at 1278.6 and 1273.9 eV, corresponding to the Tb 3d signals (Fig. 2h). Furthermore, the presence of YF$_3$:Tb$^{3+}$ was confirmed by powder X-ray diffractometry (PXRD) analysis of HNTs@YF$_3$:Tb$^{3+}$ (Supplementary Fig. 6 and Supplementary Tables 2 and 3), which exhibited patterns consistent with standard reference data. Collectively, these comprehensive characterizations conclusively confirm the successful synthesis of HNTs@YF$_3$:Tb$^{3+}$ nanoparticles.

### Structural and optical characterizations
To gain insights into the structure of HNTs and HNTs@YF$_3$:Tb$^{3+}$ nanoparticles, we employed scanning transmission electron microscopy (STEM) in conjunction with energy dispersive X-ray spectroscopy (EDX). A distinct morphology was observed for HNTs (Fig. 3a, b), which differed from that of HNTs@YF$_3$:Tb$^{3+}$ (Fig. 3c). Notably, the HNTs@YF$_3$:Tb$^{3+}$ nanoparticles exhibited an additional density attached to the HNT scaffold (Fig. 3c). The interplanar distance between adjacent lattice fringes in the attached nanoparticles (Fig. 3d), measured as $d = 0.37$ nm, closely matched the interplanar distance of YF$_3$ (011) planes. This observation strongly suggests that this density corresponds to the attachment of YF$_3$:Tb$^{3+}$ nanoparticles, further supported by the high-angle annular dark-field (HAADF) STEM images (Supplementary Figs. 7 and 8).

To validate this finding, we conducted scanning transmission electron microscope-energy dispersive X-ray (STEM-EDX) elemental mapping of HNTs@YF$_3$:Tb$^{3+}$ and observed distinct distributions of different elements (Supplementary Figs. 7–9). Signals from Al, Si, and O were found to be distributed along the HNT component, while the Y, F, and Tb signals were confined to the attached entities. This elemental mapping analysis unequivocally confirms that the HNT scaffold is decorated with YF$_3$:Tb$^{3+}$ particles, and the successful integration of Tb into the YF$_3$ lattice.

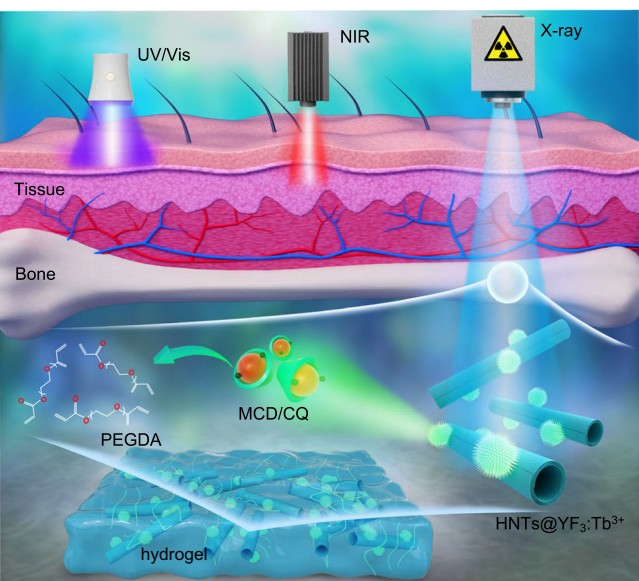

**Fig. 1 | A comparative analysis of tissue penetration depths achieved with different excitation sources.** Ultraviolet/visible (UV/Vis) light exhibits restricted tissue penetration, typically reaching only a few millimeters, thus severely hampering its utility in biological systems. Near infrared ray (NIR) light with longer wavelengths can enhance penetration depth to some extent. Nonetheless, existing photo-crosslinking methods fall short in achieving deep-tissue penetration, especially within bone structures. Compared to light, X-rays, a form of electromagnetic waves extensively utilized in biomedical applications, exhibit remarkable penetration capabilities through both bones and human tissue. Herein, we present a breakthrough X-ray-controlled polymer crosslinking system consisted of containing polyethyleneglycoldiacrylate (PEGDA), complex of camphorquinone and methyl-β-cyclodextrin (MCD/CQ), and halloysite nanotubes-based X-ray-activated visible persistent luminescent emitting phosphors (HNTs@YF$_3$:Tb$^{3+}$), referred to as Xcrosslinking, that achieves hydrogel formation within deep tissues and bone structures with advantages including highly spatial controllablity, noncontact external stimuli, and uncritical demand for the environment.

We conducted X-ray-excited luminescence emission spectroscopy to investigate the emission properties of HNTs, HNTs-EDTA, and HNTs@YF$_3$:Tb$^{3+}$. As shown in Fig. 3e, both HNTs and HNTs-EDTA exhibited no luminescence when exposed to a 30 mA 40 kV X-ray irradiator. In striking contrast, the X-ray-excited luminescence emission spectrum of HNTs@YF$_3$:Tb$^{3+}$ revealed four distinct peaks at 490, 545, 588, and 623 nm, corresponding to the $^5D_4 \rightarrow {}^7F_6$, $^5D_4 \rightarrow {}^7F_5$, $^5D_4 \rightarrow {}^7F_4$, and $^5D_4 \rightarrow {}^7F_3$ transitions, respectively[42].

To further explore the X-ray-induced long persistent luminescence properties of HNTs@YF$_3$:Tb$^{3+}$, we recorded persistent luminescence decay curves after irradiation with an X-ray irradiator for 1 min. Remarkably, the afterglow persisted for up to 50 s with gradual attenuation (Fig. 3f). The visible afterglow was clearly observable to the naked eye following X-ray irradiation (Fig. 3g) and remained visible for 1 min (Fig. 3h). Furthermore, we obtained the persistent luminescence decay curve of HNTs@YF$_3$:Tb$^{3+}$ monitored at 545 nm for varying irradiation times (20 to 600 s). As depicted in Fig. 3j, the afterglow intensity initially declined rapidly within the first 60 s, followed by a slow decay. Notably, prolonging the irradiation time led to a noticeable increase in afterglow intensity. The dependence of X-ray irradiation time (*t*) on the intensity of afterglow at 600 s was investigated. The plots between log*t* and the intensity of afterglow on 600 s were fitted linearly, in which the correlation index (R) was 0.983 (Supplementary Fig. 10).

Our comprehensive structural and optical studies highlight the outstanding potential of HNTs as a scaffold for YF$_3$:Tb$^{3+}$ attachment, resulting in a nanostructure exhibiting robust visible luminescence. Moreover, the emission from HNTs@YF$_3$:Tb$^{3+}$ did not include any UV light components under X-ray activation. This not only mitigates safety concerns associated with UV radiation in in vivo applications but also positions HNTs@YF$_3$:Tb$^{3+}$ as a viable in vivo light source for initiating the polymerization of photo-crosslinking hydrogels.

## Validation of X-ray-excited polymerization for gelatinization reaction

To assess the capacity of HNTs@YF$_3$:Tb$^{3+}$ in inducing photo-cross-linking, we combined it with camphorquinone (CQ), a widely used photoinitiator in the biomedical field known for its compatibility with the visible light region[43]. Furthermore, the absorption spectra of CQ closely overlapped with the luminescence and afterglow of HNTs@YF$_3$:Tb$^{3+}$. To enhance the photo-activity and water-solubility of CQ, we encapsulated it within the hydrophobic cavity of methyl-β-cyclodextrin (MCD). The successful formation of the complex of camphorquinone and methyl-β-cyclodextrin (MCD/CQ) was confirmed through NMR studies (Supplementary Fig. 11). Additionally, we employed a FDA-approved polyethyleneglycoldiacrylate (PEGDA)[44] as the diolefinic monomer and triethanolamine as the hydrogen donor. Triethanolamine is a commonly-used co-initiator in photo-polymerizations and has been demonstrated to show low oxidation potential and the low energy barrier[43]. Parallel experiments were conducted to investigate gel formation under different conditions (Fig. 4a–d). To simulate a deep tissue environment, we placed a 2 cm-thick animal tissue between the X-ray source and the reaction vial. Gel formation was solely observed upon X-ray exposure (30 mA, 40 kV) when all key components were present (PEGDA, MCD/CQ, triethanolamine, and HNTs@YF$_3$:Tb$^{3+}$).

To validate the formation of hydrogel under these conditions, we employed gel permeation chromatography (GPC) coupled with a refractive index (RI) detector (Supplementary Fig. 12). The rheological characteristics of the resulting hydrogel were demonstrated by the storage modulus (G′) being greater than the loss modulus (G″) over an average frequency ($\omega$) range of 1–100 rad s$^{-1}$ (Supplementary Fig. 13). Collectively, these results confirm that the combination of all required components, along with X-ray activation of HNTs@YF$_3$:Tb$^{3+}$, leads to successful gel formation (Fig. 4e, f). PEGDA with a molecular weight of 400 g mol$^{-1}$ is regarded as the optimal component for further studies because it can be dissolved in water at any proportion and is able to achieve a gelation state in a shorter time than that of a higher molecular weight.

Additional parallel experiments were also conducted by using a mono-vinyl olefin, poly(ethylene glycol) methyl ether acrylate (PEGMA), to investigate whether the polymerization behavior can take place by using these components: v- PEGMA; vi: PEGMA, MCD/CQ, and triethanolamine; vii: PEGMA and HNTs@YF$_3$:Tb$^{3+}$; viii: PEGMA, MCD/CQ, and HNTs@YF$_3$:Tb$^{3+}$; ix: PEGMA, MCD/CQ, triethanolamine, and HNTs@YF$_3$:Tb$^{3+}$. The monomer conversions (%) for the above-mentioned groups (v, vi, vii, and viii) are all calculated as ca. 0% based on the NMR analysis (Supplementary Figs. 15 and 16), indicating the polymerization cannot take place when using the above-mentioned groups containing different components. The results match well with the findings from GPC curves. The radical cannot be generated from X-ray/CQ or bypassing the composite, and the co-initiator triethanolamine is also essential for the Xcrosslinking system. Only a combination of all required components (group ix), along with X-ray activation of HNTs@YF$_3$:Tb$^{3+}$, can lead to successful free radical polymerization. The monomer conversion (%) of the group (ix) was calculated as 5.3%, 45.0%, 46.0%, and 49.0% as the irradiation time ranged from 1 min, 3 min, 5 min, and 10 min, respectively (Supplementary Fig. 17). The monomer conversion (%) increases with the increase of the irradiation time. A significant difference can be found between the monomer conversion (%) of 1 min and 3 min. After 3 min, the monomer conversion (%) changes slowly with the continuous increase of irradiation time.

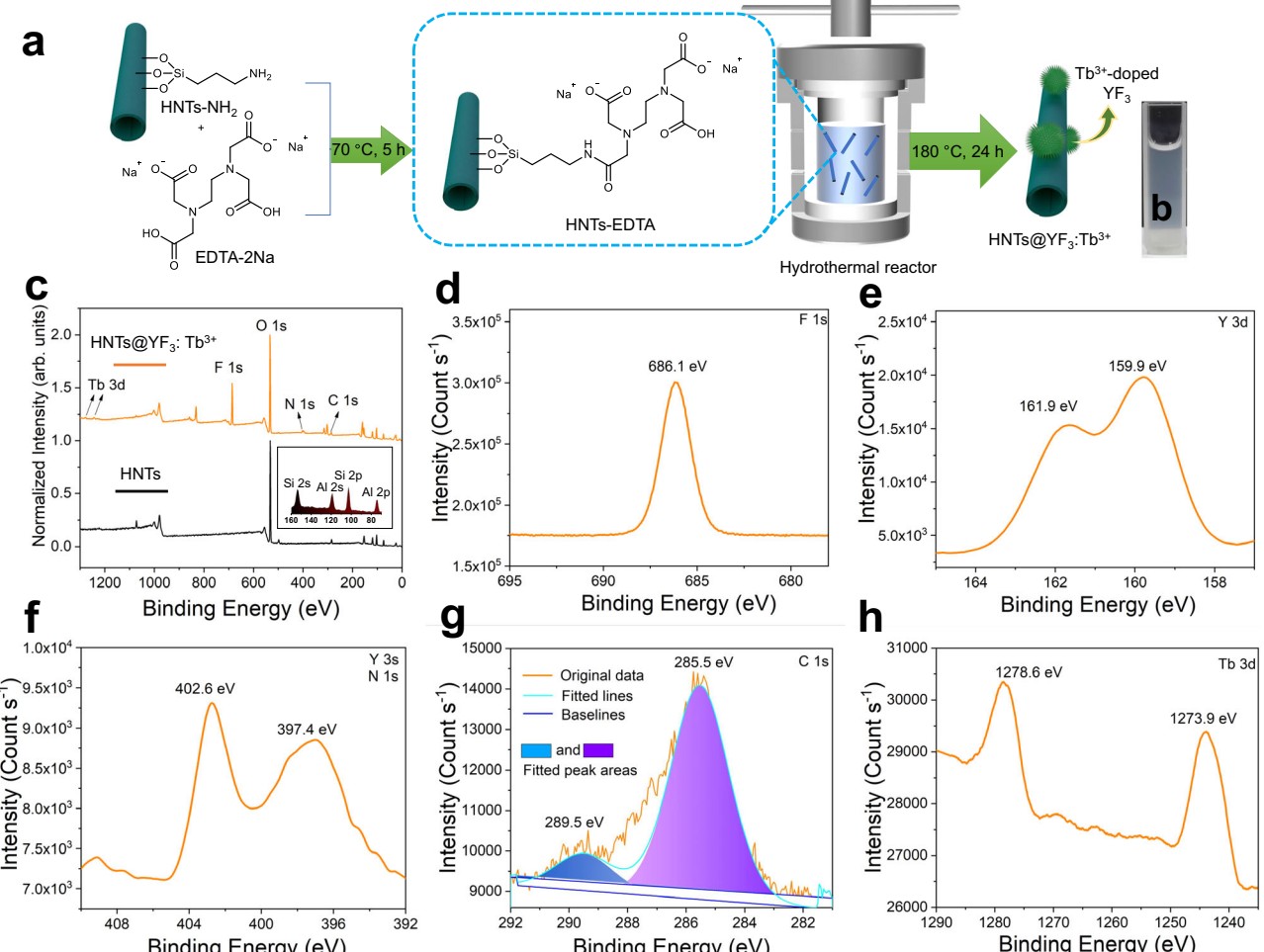

**Fig. 2 | Synthesis and characterization. a** Schematic representation of synthesis of ethylenediaminetetraacetic acid-derivatized halloysite nanotubes (HNTs-EDTA) and halloysite nanotubes-based X-ray-activated visible persistent luminescent emitting phosphors (HNTs@YF$_3$:Tb$^{3+}$). **b** Photographic image of HNTs@YF$_3$:Tb$^{3+}$ in water. **c** X-ray photoelectron spectroscopy (XPS) patterns of halloysite nanotubes and HNTs@YF$_3$:Tb$^{3+}$ in the region from 1280 to 0 eV. The inset picture is the enlarged region relating to Al and Si signals from the XPS pattern of HNTs. **d** F 1s region for HNTs@YF$_3$:Tb$^{3+}$. **e** Y 3d region for HNTs@YF$_3$:Tb$^{3+}$. **f** Y 3 s and N 1 s regions for HNTs@YF$_3$:Tb$^{3+}$. **g** C 1 s region for HNTs@YF$_3$:Tb$^{3+}$. **h** Tb 3d region for HNTs@YF$_3$:Tb$^{3+}$.

The X-ray-induced luminescence, afterglow, and gelatinization behavior have motivated us to explore a crosslinking reaction system based on X-PLNPs, which we have named Xcrosslinking. This system offers the advantage of deep tissue applicability in nontransparent systems. To minimize X-ray exposure, we adopted an on-off-on circulation approach. Leveraging the persistent afterglow of HNTs@YF$_3$:Tb$^{3+}$, we utilized it as a long-lasting light source for the Xcrosslinking system. In our experimental setup, as illustrated in Fig. 4g, a printer employing an up-bottom projection approach was employed. The AL01C II X-Ray Collimator (Type: 5234954; S. N. 7597; Tube current: 100 mA; Tube voltage: 50 kV; Energy: 5.0 mAs) was positioned on top, while a layer of animal tissue was placed beneath the X-ray source. A glass platform at the bottom of the system provided support for the samples. The solution containing PEGDA (40%, m/v), MCD/CQ (CQ: 3%, m/v), triethanolamine (6%, m/v), and HNTs@YF$_3$:Tb$^{3+}$ (5%, m/v) was loaded into an injector. The injector, movable in a horizontal manner, delivered the HNTs@YF$_3$:Tb$^{3+}$-filled solution into the mold. The irradiation interval was fixed at 60 s, as depicted in Fig. 4h. Under these conditions, we successfully fabricated self-standing pentagram-shaped hydrogels exhibiting the desired elastomeric properties (Fig. 4i–k). Notably, the obtained hydrogel demonstrated excellent structural integrity and mechanical properties even when stretched and bent to extreme angles (Fig. 4l–n).

Collectively, our findings highlight the feasibility and high-quality gel material produced through the utilization of the Xcrosslinking system. Furthermore, our experiments conducted in the presence of 2 cm thick animal tissue suggest the potential applicability of this strategy in vivo.

## Characterization of in vivo gelatinization reaction

To investigate the potential of HNTs@YF$_3$:Tb$^{3+}$ for in vivo applications, we first assessed its cytotoxicity using a (3-(4,5-dimethylthiazol-2-yl)−2,5-diphenyltetrazolium bromide) (MTT) assay in mouse fibroblast L929 cells. Remarkably, HNTs@YF$_3$:Tb$^{3+}$ demonstrated negligible biological toxicity, as cell viability remains above 90% even at relatively high concentrations (1000 μg mL$^{-1}$) after 24 h, 48 h, and 7 days of treatment (Fig. 5a). The HNTs@YF$_3$:Tb$^{3+}$-treated L929 cells with the exposure to X-ray (1.5 mGy) also exhibited high cell viability (>90%). Moreover, we also evaluated the impact of green light emitted by light-emitting diode (LED) on cell viability. A good cell viability (>90%) can also be achieved. The results indicate that the obtained HNTs@YF$_3$:Tb$^{3+}$, 1.5 mGy X-ray, and the generated green light exhibit negligible biological toxicity on mouse fibroblast L929 cells. Encouraged by these favorable in vitro results, we proceed to evaluate the system in a Sprague Dawley rat model (Fig. 5b).

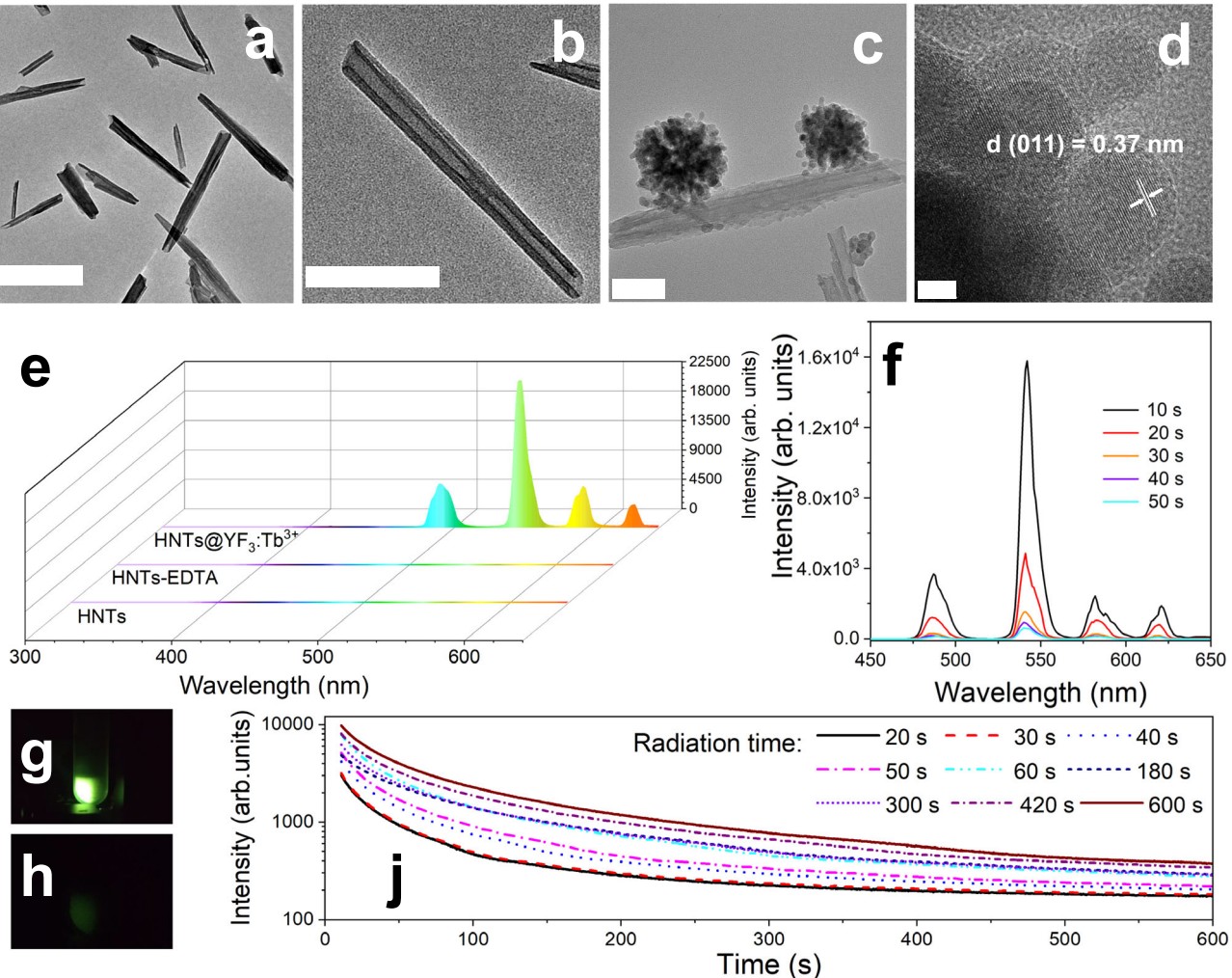

**Fig. 3 | Micromorphological and optical characterizations. a** Transmission electron microscope (TEM) image of pristine halloysite nanotubes (HNTs), scale bar: 500 nm. **b** TEM image of pristine HNTs, scale bar: 200 nm. **c** TEM image of halloysite nanotubes-based X-ray-activated visible persistent luminescent emitting phosphors (HNTs@YF$_3$:Tb$^{3+}$), scale bar: 100 nm. **d** High resolution transmission electron microscope (HETEM) image of HNTs@YF$_3$:Tb$^{3+}$, scale bar: 5 nm (Each experiment in **a-d** was repeated for 3 times independently with similar results). **e** X-ray-excited luminescence emission spectra of HNTs, ethylenediaminetetraacetic acid-derivatized halloysite nanotubes (HNTs-EDTA), and HNTs@YF$_3$:Tb$^{3+}$. The emission peaks at 490, 545, 588, and 623 nm that are assigned to the $^5D_4 \rightarrow ^7F_6$, $^5D_4 \rightarrow ^7F_5$, $^5D_4 \rightarrow ^7F_4$, and $^5D_4 \rightarrow ^7F_3$ transitions, respectively. **f** Afterglow spectra of HNTs@YF$_3$:Tb$^{3+}$ recorded at 10 s, 20 s, 30 s, 40 s, and 50 s of afterglow attenuation. **g** Photographic image of HNTs@YF$_3$:Tb$^{3+}$ taken under X-ray; **h** Photographic image of HNTs@YF$_3$:Tb$^{3+}$ taken at 1 min afterglow time after irradiation with X-rays for 1 min. **j** Afterglow intensity from HNTs@YF$_3$:Tb$^{3+}$ monitored at 545 nm as a function of time.

In clinical settings, X-ray irradiation is typically limited to durations of <1 s. Leveraging the optical properties of HNTs@YF$_3$:Tb$^{3+}$, we hypothesized that these nanoparticles can serve as a rechargeable persistent light source for in vivo gelatinization under clinically safe conditions. To validate this hypothesis, we randomly divided Sprague Dawley rats into three groups: (i) the negative control group without any treatment (Fig. 5c), (ii) the control group treated with HNTs@YF$_3$:Tb$^{3+}$ suspension (10%, m/v) containing PEGDA, MCD/CQ, and triethanolamine without exposure to X-rays (Fig. 5d), and (iii) the test group treated with HNTs@YF$_3$:Tb$^{3+}$ suspension (10%, m/v) containing PEGDA, MCD/CQ, and triethanolamine, and exposed to X-rays (Fig. 5e). For rats receiving X-ray irradiation (group iii), the back area of the rat was irradiated at a dose equivalent to the routine imaging dose for human fingers. The irradiation was repeated ten times, with 60 s intervals between each repetition, resulting in a total dose of 1.5 mGy, well within the limits of typical clinical X-ray applications.

Figure 5i–k showcase the dissected backs of the rates. We observed the formation of hydrogel exclusively in the test groups, as indicated by the presence of a yellow gel (group iii), whereas the control groups did not exhibit any gel formation (group ii showed the presence of the applied solution, but no gel). Furthermore, histomorphological analysis of the skin tissue conducted 7 days post-treatment reveal minimal differences between the control and test animals, with no evidence of inflammatory phases (Fig. 5i–k and Supplementary Fig. 19). This suggests the excellent biocompatibility of the injected substances and the resulting hydrogels.

To gain further insight into the process, we confirmed the presence of the injected HNTs@YF$_3$:Tb$^{3+}$ suspension in the X-ray images (Fig. 5l). Notably, due to the strong contrast effect from HNTs@YF$_3$:Tb$^{3+}$, we were unable to directly track the gelatinization process based on time-resolved X-ray films (Supplementary Fig. 20). To verify that the yellow colloidal structure indeed corresponds to a gel, we isolated this substance from the animal and subjected it to rheological investigation. Our findings demonstrate that the obtained yellow colloidal substance exhibits typical hydrogel properties (Fig. 5m).

Overall, our results establish that the luminescence and afterglow emitted by HNTs@YF$_3$:Tb$^{3+}$ enable the formation of photo-

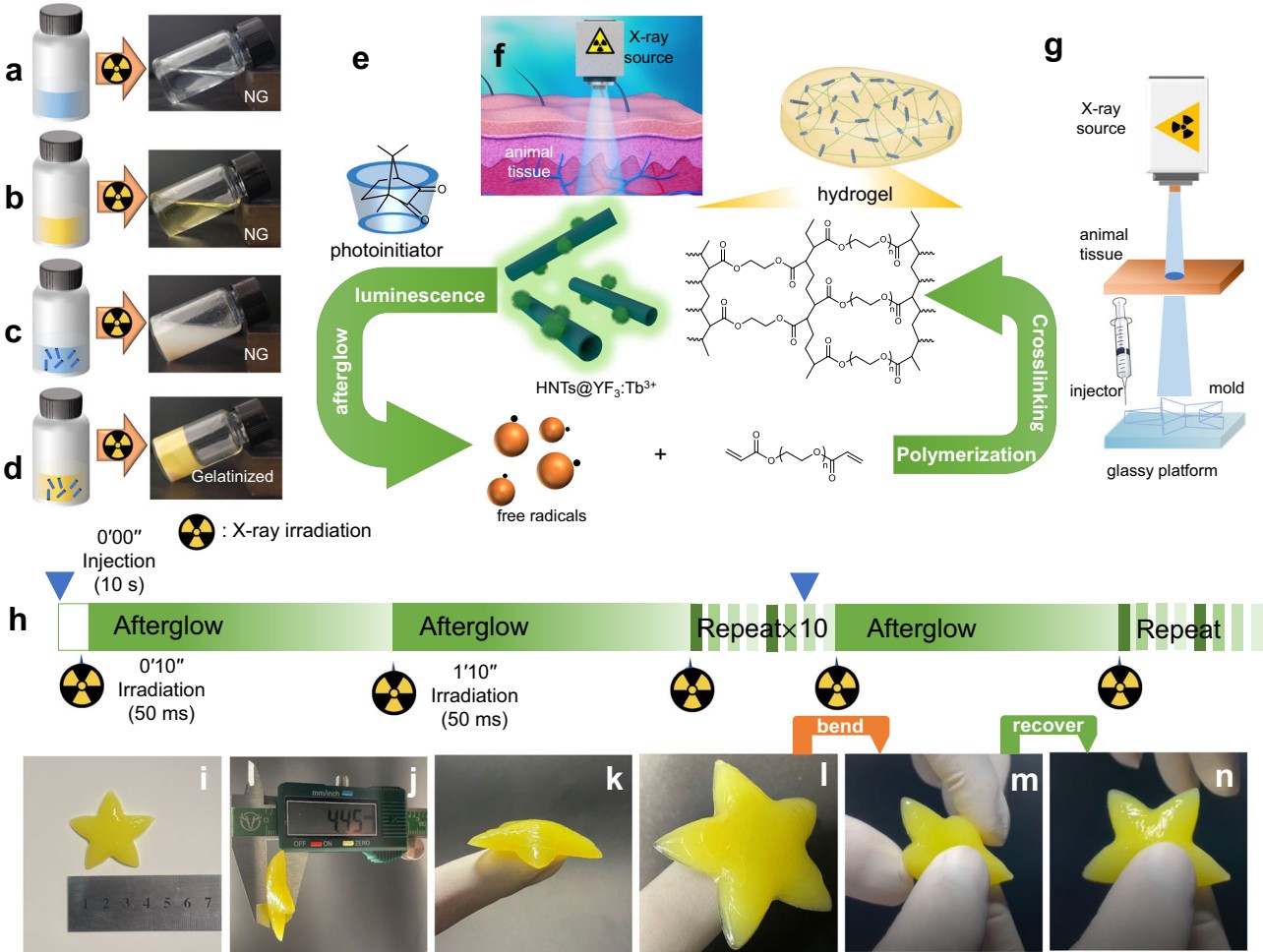

**Fig. 4 | Gelatinization conditions and hydrogel properties. a** Photographic image depicting gelatinization behavior of the solution containing poly-ethyleneglycoldiacrylate (PEGDA) in parallel experiment (NG is abbreviated from not gelatinized). **b** Photographic image depicting gelatinization behavior of the solution containing PEGDA, complex of camphorquinone and methyl-β-cyclodextrin (MCD/CQ), and triethanolamine in parallel experiment. **c** Photographic image depicting gelatinization behavior of the solution containing PEGDA and halloysite nanotubes-based X-ray-activated visible persistent luminescent emitting phosphors (HNTs@YF₃:Tb³⁺) in parallel experiment. **d** Photographic image depicting gelatinization behavior of the solution containing PEGDA, MCD/CQ, triethanolamine, and HNTs@YF$_3$:Tb³⁺ in parallel experiment. **e** Polymerization and crosslinking mechanism in the Xcrosslinking system. **f** Illustration of the X-ray source used in gelatinization studies. **g** Illustration of the printing apparatus utilizing a pentagram-shaped mold with dimensions of ~4.2 cm (length) and 0.45 cm (height). **h** Schematic representation of the on-off-on circulation process employing X-ray as the energy source. The irradiation interval was fixed at 60 s. **i**–**k** Photographs showcasing the prepared hydrogel. **l**–**n** Demonstration of the bending and recovering behaviors exhibited by the obtained hydrogel.

crosslinking hydrogels within biological systems under clinically safe X-ray irradiation conditions.

## On-demand gelatinization inside the bone

The remarkable tissue penetration ability of X-rays has motivated us to investigate the potential of our system within bone tissues, which holds significant promise for the treatment of conditions such as osteoporosis—a prevalent condition characterized by systemic bone loss affecting millions of individuals worldwide. To examine the capacity of our system for bone repair, we created a bone defect model by removing the marrow from a bovine bone. Subsequently, we injected a solution containing PEGDA (40%, m/v), MCD/CQ (CQ: 3%, m/v), triethanolamine (6%, m/v), and HNTs@YF₃:Tb³⁺ (5%, m/v) into the defect region (Fig. 6a illustrates the schematic procedure). Following X-ray exposure (30 mA, 40 kV, for 20 min), we retrieved the radiated bone specimen (Fig. 6b).

Remarkably, we observed the formation of a hydrogel within the bone (Fig. 6c), which can be readily separated from the bone tissue (Fig. 6d). The isolated hydrogel exhibited the expected rheological properties, with G′ > G″ as the average frequency (ω) ranged from 0.1 –

100 rad s⁻¹ (Fig. 6e). Importantly, the measured bone thickness exceeded 7.0 mm (Fig. 6f) on each side. Collectively, these findings highlight the capability of our approach to enable the formation of photo-crosslinking hydrogels within thick bone tissue.

Noted that in our research, we employed a thick, dense bovine bone model, fully polymerizing the entire cavity, to provide an illustrative example. Bovine bone, known for its higher density and stronger X-ray attenuation compared to human bones, was chosen to represent a challenging scenario. In practical medical applications, bone defects are often located in more superficial layers, and the affected area is typically smaller than our model[45,46]. Therefore, the X-ray dose required in real-world applications be lower than that used in this conceptual study.

In summary, our study presents a pioneering Xcrosslinking strategy that overcomes the limitations of current methods by enabling the formation of photo-crosslinked hydrogels within bone tissue. This breakthrough is achieved by leveraging X-PLNPs as a persistent light source, which are immobilized onto HNTs to create a biocompatible and safe system with uniform size and excellent water dispersity. Upon low-dose X-ray irradiation, the resulting

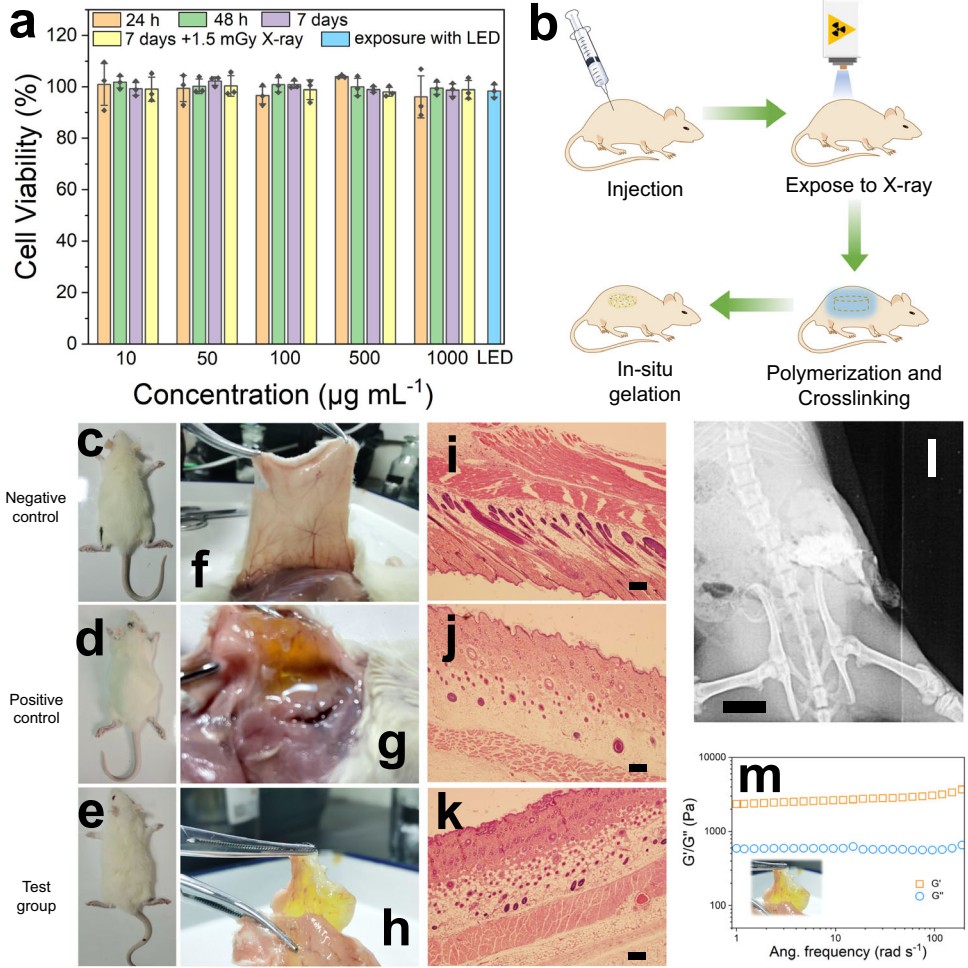

**Fig. 5 | Cytotoxicity and in vivo gelatinization studies. a** Cell viability data in mouse fibroblast L929 cells (*n* = 3). Data are presented as mean values +/- standard deviation (SD). LED is abbreviated from light-emitting diode. No significant difference was found between the control group and test groups. *P*-values for the group of 24 h (left to right): 1.000, 1.000, 0.953, 0.906, and 0.916; *P*-values for the group of 48 h (left to right): 0.976, 1.000, 1.000, 1.000, and 1.000; *P*-values for the group of 7 days (left to right): 0.995, 0.778, 0.995, 0.985, and 0.966; *P*-values for the group of 168 h + X-ray (left to right): 1.000, 1.000, 0.999, 0.983, and 0.999; *P*-values for LED group: 0.546. **b** Schematic diagram of the in vivo gelatinization process using X-ray irradiation to the back area (Tube current: 100 mA; Tube voltage: 50 kV; Exposure time: 50 m Sec; Energy: 5.0 mAs; Field of view (FOV): 8.0 cm ×10.0 cm; Height: 1.0 m). **c** Photographic image of the rat in negative control group before dissection. **d** Photographic image of the rat in positive control group before dissection. **e** Photographic image of the rat in test group before dissection. **f** Photographic image of the rat in negative control group after dissection. **g** Photographic image of the rat in positive control group after dissection. **h** Photographic image of the rat in test control group after dissection. **i** Hematoxylin & eosin (H&E)-stained skin tissues from rat in negative control group, scale bar: 500 µm. **j** H&E-stained skin tissues from rat in positive control group, scale bar: 500 µm. **k** H&E-stained skin tissues from rat in test group, scale bar: 500 µm (Each experiment in **i–k** was repeated for 3 times independently with similar results). **l** X-ray film of living rats after injection of halloysite nanotubes-based X-ray-activated visible persistent luminescent emitting phosphors (HNTs@YF$_3$:Tb$^{3+}$) suspension, scale bar: 1 cm. **m** Rheological property of the as-formed hydrogel. The inset picture is from the photographic image of rat in test control group after dissection.

HNTs@YF$_3$:Tb$^{3+}$ nanomaterial exhibits strong green luminescence and afterglow, effectively initiating deep tissue gelatinization. Importantly, we demonstrate the compatibility of our strategy within thick bone structures, achieving remarkable deep-tissue penetration in bone tissues with a thickness exceeding 7 mm.

This work represents a significant advancement in the field, introducing a transformative approach for achieving deep-tissue and bone-penetrating photo-crosslinking. The utilization of X-PLNPs as a persistent light source offers opportunities for precise and controlled hydrogel formation, overcoming the limitations of current methods. The successful implementation of our strategy opens up avenues for future clinical applications, particularly in the fields of tissue repair, regenerative medicine, and bone tissue engineering. Our system demonstrates potential for deep-tissue and bone-penetrating photo-crosslinking polymerization. Moreover, the potential of this method extends beyond bone applications. It is also applicable in non-medical

scenarios, such as repairing deep defects in materials like concretes, alloys, and composites, where the current light based method is limited[47]. These findings have broad implications for advancements in various chemical and biomedical applications. Further studies will focus on applying the proposed Xcrosslinking system to specific application scenarios by synthesizing more types of X-PLNPs with diverse radioluminescence properties. Potential application scenarios (Fig. 7) are summarized as follows: Noninvasive 2D or 3D printing is expected to repair defective or damaged deep tissues in situ in a spatially controlled manner, which can get rid of surgical operations. X-PLNPs can be excited by X-ray before being injecting into the defect and the generated luminescence can be used as a nanolight source to trigger the curing process of the photosensitive resins for dental repairs, which is beneficial to break the depth limitation of the visible light-cured dental resin, shorten the mouth opening time, and maybe a promising supplementary technique to visible light-cured dental resin

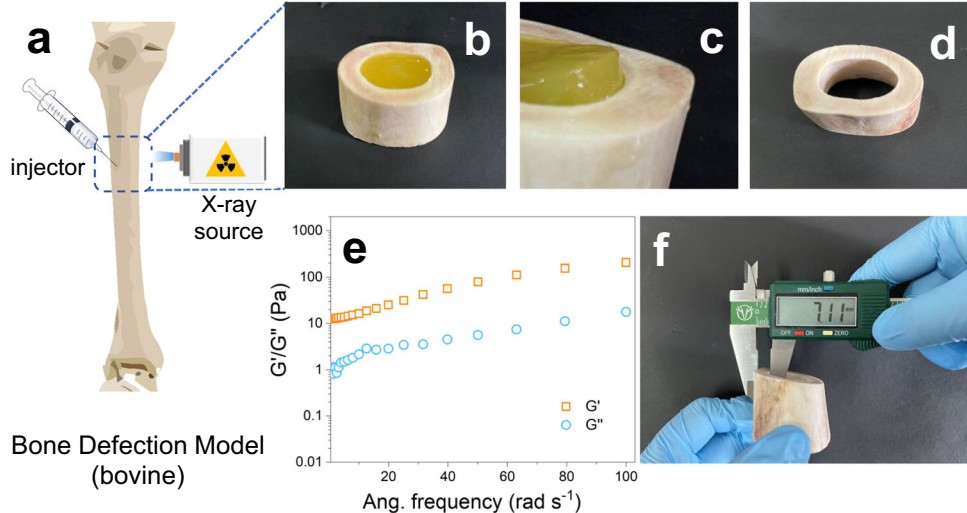

**Fig. 6 | Gelatinization studies in bone defect model. a** Schematic diagram of the model. Solution in injector contains polyethyleneglycoldiacrylate (PEGDA) (40%, m/v), complex of camphorquinone and methyl-β-cyclodextrin (MCD/CQ) (CQ: 3%, m/v), triethanolamine (6%, m/v), and halloysite nanotubes-based X-ray-activated visible persistent luminescent emitting phosphors (HNTs@YF$_3$:Tb$^{3+}$, 5%, m/v). **b** Sawed section of the bone defect model after irradiation. **c** The formed removable substance inside the bone. **d** The sawed section after removing the hydrogel. **e** Rheological property of the as-formed hydrogel. **f** Thickness test.

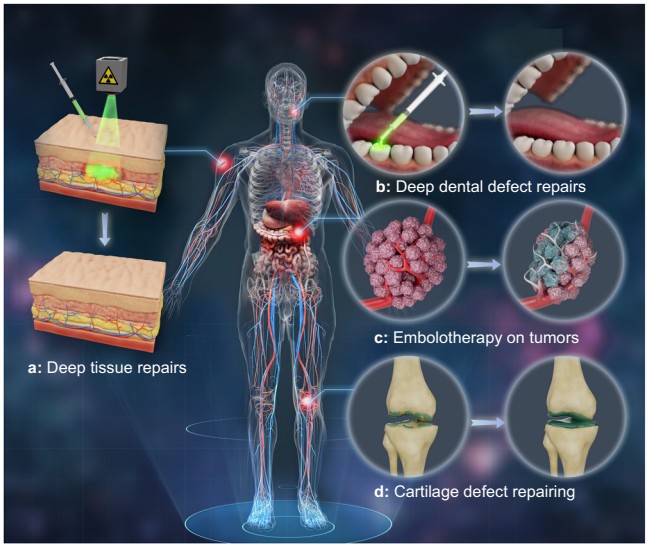

**Fig. 7 | Summary of the potential application scenarios of the Xcrosslinking system. a** Deep tissue repairs. **b** Deep dental defect repairs. **c** Emnolotherapy on tumors. **d** Cartilage defect repairs.

for repairing deep defects. The Xcrosslinking system is expected to be used in endovascular embolotherapy on malignant tumors. Following a precise interventional injection on the tumor sites and exposure to X-ray, gelatinization can take place inside the tumor vessels. The gelatinization can achieve an embolotherapy effect and thereby effectively suppress the cancerous growth. Cartilage defect repairing may also be achieved by the proposed X-crosslinking system. The crosslinking degree of swelling behavior can be controlled by changing the irradiation time and X-ray doses to meet the requirement of the mechanical properties for cartilage use.

## Methods

### Ethical statement

The protocol about the animal experiments in this study has been reviewed and approved by the Animal Ethical and Welfare Committee of Hebei University (Approval No. IACUC-2021XG008, Data: Mar. 11$^{st}$ 2021).

### Synthesis of ethylenediaminetetraacetic acid-derivatized halloysite nanotubes

The HNTs-NH$_2$ was prepared by treating HNTs (22.0 g) and anhydrous Na$_2$CO$_3$ (0.60 g, 5.7 mmol) in KH550® (90 mL). The mixture was stirred and heated at 80 °C for 48 h. HNTs-NH$_2$ was afforded by collecting the residue after certification. 1.08 g dried HNTs-NH$_2$ was added into the water solution containing 1.35 g EDTA-2Na. The nanosuspension was then transferred into a 150 mL single neck flask and heated at 70 °C with tensely magnetic stirring for 5 h. After cooling to room temperature, the residue was collected by centrifugation at 10,840 g and then washed by ethanol and acetone for three cycles. The obtained precipitation was frozen via liquid nitrogen and then dried in vacuum. The product HNTs-EDTA was obtained as white solid.

### Synthesis of halloysite nanotubes-based X-ray-activated visible persistent luminescent emitting phosphors

1.70 g HNTs-EDTA was accurately weighed and added to the aqueous solution containing Y(NO$_3$)$_3$·6H$_2$O and Tb(NO$_3$)$_3$·6H$_2$O with the molar ratio of 1: 0.9: 0.1. The system was stirred in low level for 30 min to achieve a homogeneous suspension. 1.48 g NH$_4$F was added into the system and then the pH value was adjusted to 5.0 by nitric acid. The agitation was continued for another 30 min. The obtained suspension was subjected into hydrothermal reactor with a filling rate of 70% and then heated at 180 °C for 24 h. After cooling to room temperature, the residue was collected by centrifugation at 10,840 g and then washed by ethanol and acetone for three cycles. The obtained precipitation was frozen via liquid nitrogen and then dried in vacuum. The product HNTs@YF$_3$:Tb$^{3+}$ was obtained as white solid.

### Preparation of photoinitiator

Briefly, excess solid camphorquinone was added to methyl-β-cyclodextrin methanol solutions, followed by the suspension was stirred and sonicated at room temperature for 0.5 h. Then the solution was stirred for another 24 h. After filtered through a 0.45 µm membrane, the resulting solution was frozen-dried to give the faint yellow solid product MCD/CQ.

### In vitro gelatinization study

HNTs@YF$_3$:Tb$^{3+}$ was suspended in aqueous solution (5%, m/v) containing PEGDA (40%, m/v), MCD/CQ (CQ: 3%, m/v), and triethanolamine (6%, m/v). The mixture was sealed in glassy vials and placed on

the platform in front of an X-ray source (40 kV, 30 mA). A 2.0 cm animal tissue was placed between the X-ray source and vial. The total irradiation duration is 10 min.

## Cell culture

Human cervical cancer cells (HeLa) and mouse fibroblast L929 cells were used in this study, which were purchased from Chinese Academy of Medical Sciences, Peking Union Medical College (catalog number: 1101HUM-PUMCO00011) and the Cell Bank of the Shanghai Chinese Academy of Science (catalog number: BFN60805937), respectively. The cells were maintained in Dulbecco's modified Eagle's medium (DMEM) and supplemented with 10% fetal bovine serum (FBS) within a humidified environment (37 °C) containing 5% $CO_2$ and 1% penicillin/streptomycin antibiotics.

## Cytotoxicity study

HeLa cells were seeded in 96-wellplate ($2 \times 10^4$ cells mL$^{-1}$) and then incubated with HNTs@YF$_3$:Tb$^{3+}$ at 37 °C with the concertation ranging from 10 to 1000 µg mL$^{-1}$. After 12, 24 and 48 h, MTT solution (20 µL, 5 mg mL$^{-1}$) was added to each well, another incubation process was continued for 4 h. Then the medium was removed and 100 µL DMSO was added. After homogenizing well, the absorbance of each well was measured at 490 nm by the ELX-800 microplate reader (ELISA Reader). Wells without the addition of samples were used as blank control. The cell viability (%) was calculated by the absorbance percentage of test to control. Exact P-value was calculated based on the one-way ANOVA Tukey's multiple comparisons test.

The cell viability of HNTs@YF$_3$:Tb$^{3+}$ with the concertation ranging from 10 to 1000 µg mL$^{-1}$ for 24 h, 48 h and 7 days was evaluated using mouse fibroblast L929 cells following a similar MTT method to the former case. The effect of X-ray radiation on cell viability was investigated by placing the cell-filled 96-wellplate with the addition of HNTs@YF$_3$:Tb$^{3+}$ with the concertation ranging from 10 to 1000 µg mL$^{-1}$ under the AL01C II X-Ray Collimator (Type: 5234954; S. N. 7597; Tube current: 100 mA; Tube voltage: 50 kV; Exposure time: 50 mSec; Energy: 5.0 mAs; FOV: 8.0 cm × 10.0 cm; Height: 1.0 m) at 37 °C with a total dose of 1.5 mGy, identical to that of in vivo gelatinization studies. Then the 96-wellplate were treated as usual for 7 days. To examine the effect of visible light on cell viability, the cell-filled 96-wellplate was exposed to green LED lamp for 1 h, which can emit light with obviously higher intensity than the luminescence generated by HNTs@YF$_3$:Tb$^{3+}$ under X-ray. Then the 96-wellplate were also treated as usual for 7 days. The control groups were isolated from the X-ray or LED light by a lead plate. The cell viability was examined following the same MTT method as Hela cells. Exact P-value was also calculated based on the one-way ANOVA Tukey's multiple comparisons test.

## In vivo gelatinization study

Sprague-Dawley male rats (6–8 weeks old) were purchased with protocols approved from the Experimental Animal Center of Hebei Province, Shijiazhuang, China. The protocol has been reviewed and approved by the Animal Ethical and Welfare Committee of Hebei University (Approval No. IACUC-2021XG008, Data: Mar. 11st 2021). Sprague-Dawley rats were fed with regular food and enough water. They were housed in standard plastic cages and place under a certain temperature (25 ± 2ºC). The Sprague-Dawley rats were randomly divided into negative control group (n = 3), positive control group (n = 3) and test group (n = 3). All the rats were fasted for 12 h prior to experiments. The rats in negative control group were kept as usual without any other treatment. For the positive control group, the HNTs@YF$_3$:Tb$^{3+}$ was suspended in aqueous solution containing PEGDA, MCD/CQ, and triethanolamine and then injected into the back of the rats (5 mL kg$^{-1}$). For test group, the HNTs@YF$_3$: Tb$^{3+}$ suspension was also injected into to rats following the same manner. The rats were fastened on the wooden rat board and then were exposed to X-ray via a

Collimalor AL01C II X-Ray Collimator. Then the rats were treated as usual. After 30 min, one of the rats from each group was sacrificed by anesthetic and dissected. The injection sites were carefully investigated. Other rats in the positive control group and test group were left intact. On day 7, the rats were all sacrificed by anesthetic and dissected. The skin tissue from the injection sites were harvested, fixed in 10% formalin, processed routinely into paraffin, stained with hematoxylin and eosin (H&E) and pathology are examined by an Olympus DP 72 microscope camera.

## Bone defect model

One piece of bovine bone, purchased from Alibaba Co. Ltd., was used to establish the bone defect model. The bone was cleaned and sawed into two pieces in the middle, one of which was subjected to further use and immersed in deionized water for 30 min. The internal tissues in the opening site were removed by drill and tweezer to afford a hole with a depth of ca. 5 cm with a pinhole of ca. 2 mm at the side of the hole. The treated bone was carefully washed by deionized water. Then the top of the hole was sealed with a lead plate to afford a bone defect model.

## Reporting summary

Further information on research design is available in the Nature Portfolio Reporting Summary linked to this article.

## Data availability

The data generated in this study are provided in Supplementary Information/Source Data file. The full image dataset is available from the corresponding author upon request. Source data are provided with this paper.

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

## Acknowledgements

Hailei Zhang acknowledges the financially support from the National Natural Science Foundation of China (No. 22102045), Central Guidance on Local Science and Technology Development Fund of Hebei Province (No. 226Z1301G), the President Foundation of Hebei University (No. XZJJ202210), and the Key Project from Natural Science Foundation of Hebei Province (No. B2020201072). Y.Y acknowledges the support from the National Natural Science Foundation of China (No. 11974097). G.H and K.H. were exclusively supported by University of Massachusetts. Hailei Zhang greatly thanks to Prof. Xinwu Ba for his valuable suggestions for this paper.

## Author contributions

Hailei Zhang conducted the majority of experiments, ran the data analysis, conceived part of the project and wrote the original manuscript. B.T. ran a portion of synthesis experiments. B.Z. ran the characterization of nanoparticles. K.H. ran the data analysis. S.L. and Y.Z. performed the additional tests in revision stage. Haisong Zhang is responsible for animal studies. L.B. contributed formal analysis. Y.W. ran a portion of synthesis experiments. Y.C. is responsible for cell analysis. Y.Y. conceived the reaction route and supervised the project. G.H. conceptualized, conceived and supervised the project and experiments and wrote the manuscript

## Competing interests

The authors declare no competing interests.
