## [Peer Review File · Nature Communications]

X-ray-activated polymerization expanding the frontiers of deep-tissue hydrogel formationREVIEWER COMMENTS

Reviewer #1 (Remarks to the Author):

Zhang et al reported a unique x-ray triggered crosslinking polymerization. The mechanism is utilizing X-ray to activate a visible persistent luminescent and transfer emission energy to radical photoinitiator, which will generate radicals for the further crosslinking polymerization of di-vinyl monomers. Due to deep penetration of x-ray, the authors demonstrated this crosslinking polymerization can happen in deep tissue and bone tissue, which is far superior than other UV/vis or NIR photopolymerization methods. This is a very interesting study. It will obviously attract the interest from polymer scientists and biomedical engineers. It will bring significant value to the research community. Also, the synthesis of the composite and in-vitro and in-vivo experiments for hydrogel formation using photo-crosslinking polymerization are well designed and sufficient to support the conclusion. The only question and concern from the current reviewer is the polymerization efficiency.

Major concern:

1. Given this is a process of radical polymerization and the radical is generated by X-ray activation, the efficiency of the X-ray activated polymerization should be measured by the polymerization rate of the monomers, or radical generation rate. The dosing of the composite (>5%, w/v) and CQ photoinitiator (3%) are relatively high compared to most visible light triggered photocatalytical polymerization systems (<0.1% or even ppm photocatalysts). The polymerization rate can be evaluated by monomer conversion versus irradiation time, in which monomer conversion can be easily measured by ¹HNMR. However, the monomer should be a mono-vinyl olefin (such as PEG acrylate) instead of di-vinyl. In the current study, the authors used di-vinyl PEGDA to make hydrogel which will be difficult to measure the monomer conversion. Even <5% monomer was converted, the solution at 40% PEGDA will still gel up. The low G' and G'' (< 1kPa) for all the hydrogels may suggest very low monomer conversions. Therefore, the authors should give a sense how efficient the radical will generate (or how fast the polymerization will be) from this x-ray activated process.
2. The control experiments in Figure 3A to demonstrate necessity of all four components need more convincing data from ¹HNMR or others. The GPC (Figure S9) is known not sensitive to low amount of impurity in polymers. This data is important because the authors should rule out other pathways for radical generation, such as direct radical generation from X-ray/CQ and bypassing the composite.

Minor comments:

1. Figure 2c: the scale bar is not clear enough, which shows 100 nm in the image but was claimed as 200nm.
2. Figure 2F: 'HNTs@NaYF₄:Tb³⁺' or 'HNTs@YF₃:Tb³⁺', is it a typo? It also appears in supporting information four times.
3. 'm/v' and 'W/v' are both used in the paper, which should be maintained consistent.
4. The composite materials were used differently in in-vitro (5% suspension) and in-vivo (10%) cases. What is the concern to use different amounts?
5. Page 14, line 409, Experimental section 3.2: '10 min' is the total irradiation duration or total gelation processing time?
6. What is the standard in Figure S9? Are there any peaks at high MW area (0-6 min elution time)? Full spectra for all three samples should be provided.

Reviewer #2 (Remarks to the Author):

The manuscript still needs improvement.

Note from the Editor: the report of Reviewer #2 is attached.

Reviewer #3 (Remarks to the Author):

To address the limitations of current photo-crosslinking methods in achieving deep-tissue penetration, this study designed a X-ray-controlled polymer cross-linking method utilizing HNTs@YF₃:Tb³⁺ as a visible luminescent phosphor. This study to some extent expanded the use of hydrogels in biomedical applications. Although the findings are interesting, there are some shortcomings in presenting the significance and novelty of this work. Some data should be further analyzed and even more data should be supplemented to improve this manuscript. The following issues need to be clarified.

1. The author only presented the gelatinization mechanism and environment of the hydrogel, without delving into specific application scenarios. Consequently, the scope of its applicability remains unclear, hindering a comprehensive evaluation of its clinical significance. It is suggested to supplement specific application scenarios.
2. Currently, there exist several mechanisms to confer injectability upon hydrogels and facilitate their gelation within deep tissues. Examples include dynamic covalent hydrogels, temperature-sensitive hydrogels, shear-thinning hydrogels, and more. These hydrogels can undergo gelation without relying on external devices to initiate cross-linking after injection. In comparison to the aforementioned hydrogels, what are the distinct advantages of X-ray-activated hydrogels in medical applications? The authors should provide a more specific discussion to elucidate the innovative aspects of this study.
3. The statement "Our results demonstrated high biocompatibility and the potential safety of this approach, as it enabled depth-independent in situ crosslinking activation." in the introduction raises the question of whether the cross-linking system employed in this study is truly unaffected by tissue depth. Substantiating this conclusion necessitates additional experimental evidence.
4. Please provide further details regarding the rationale behind selecting HNTs as the substrate in this study and elucidate their unique advantages over other materials. Additionally, please illustrate the decision to employ a hollow structure rather than a solid nanorod, and discuss the implications of this choice on the overall impact of the study.
5. The authors should explore the impact of PEGDA's molecular weight and content on both gelation time and rheological properties of the hydrogel system. Furthermore, it is essential to determine whether PEGDA with a molecular weight of 400 represents the optimal component.
6. The author should supplement additional experiments to elucidate the swelling behavior of the hydrogel following irradiation-induced gelatinization, as the utilization of these hydrogels in vivo entails exposure to a significant volume of bodily fluids.
7. Please confirm whether there are unreacted monomers following the gelation of the reaction stock solution in vivo. Will the hydrogel degrade in vivo? Moreover, are there any potential adverse effects associated with long-term retention of the hydrogel in the body, particularly within the bones?
8. Although gelatinization within the bone is achievable, it requires an extended exposure time under X-ray radiation. However, it remains unclear whether this exposure condition is within the acceptable range of safety. The mention of "low-dose X-ray irradiation" lacks specificity and requires further clarification. Please provide the safe dose for X-ray irradiation and present the corresponding results and discussions on the gelation ability of the system in bone under the safe irradiation dose.
9. The article stated "The author recorded persistent luminescence decay curves after irradiation with an X-ray irradiator for 1 minute. Remarkably, the afterglow persisted for up to 50 seconds with gradual attenuation. Prolonging the irradiation time led to a noticeable increase in afterglow intensity." It remains unclear whether there is a linear relationship between X-ray irradiation time and the duration and intensity of afterglow. It is important to investigate whether adjusting the X-ray irradiation time can accurately control the luminescence of deep internal light sources and further modulate the degree of hydrogel crosslinking.
10. The examination of the effect of HNTs@YF₃:Tb³⁺ on cell proliferation limited to a 48-hour period lacks sufficient persuasive power, necessitating an extension of the research time to a minimum of 7 days. Furthermore, in the subcutaneous gelatinization experiment, while presenting low-magnification images of rat skin tissues, it is essential to supplement high-magnification images of the tissue adjacent to the hydrogel to more clearly assess the inflammatory response caused by the material. Additionally, solely evaluating the biocompatibility of HNTs@YF₃:Tb³⁺ is not enough. As previously mentioned, other existing photo-crosslinking methods may pose DNA damage risks to cells, thus raising concerns regarding safety. Therefore, it should be worthwhile for this study to investigate the effects of X-ray radiation and stimulated visible light on cell viability.

Reviewer comments:

The study by Zhang *et al.* presented a novel approach to hydrogel formation using low-dose X-ray-activated persistent luminescent phosphor through in situ photo-crosslinking activation. The proposed HNTs@YF3:Tb³⁺ nanocomposites demonstrated excellent water dispersibility and X-ray-excited luminescence. Hydrogel-forming capability of the HNTs@YF3:Tb³⁺ upon X-ray was confirmed through *in vitro* and *in vivo* experiments. However, the manuscript lacks adequate explanations and validations for key scientific inquiries, and the presented data are insufficient to substantiate the proposed mechanisms and relevant conclusions in the research. Therefore, a major revision of the manuscript is necessary.

1. The introduction did not sufficiently highlight the innovation and superiority of this work. While previous researchers have reported *in situ* hydrogel formation using X-ray irradiation-induced polymerization (e.g., *Soft Matter*, 2012, 8(5): 1420-1427), the concept of "Xcrosslinking" proposed in this study was meaningful. The authors are encouraged to clarify the innovation and superiority of their work.

2. Safety evaluation upon low-dose X-ray exposure with 30 mA and 40 kV (in line 219) and a comparison with typical clinical X-ray parameters (in line 286) should be addressed. Furthermore, selection of HeLa cells for assessing HNTs@YF3:Tb³⁺ cytotoxicity should be justified. Additionally, toxicity evaluation of HNTs@YF3:Tb³⁺ upon X-ray exposure and its impact on normal cells need to be discussed.

3. The manuscript lacked in-depth discussions. For instance, NMR characterization towards MCD & MCD/CQ in section 2.2 and experimental data to confirm gelation formation in sections 2.3 and 2.5 were insufficiently described. Furthermore, rationale behind choosing triethanolamine as the

hydrogen donor and mechanism of the free radical and hydrogel formations with X-ray radiation require detailed explanations and relevant experimental validations.

4. Numerous formatting and mistake issues were found in the manuscript, such as figure caption format in Scheme Figure & Figure 5F, missing space between value and unit in Figure 1G, lacking "Figure S5-S7" in appropriate positions within lines 161~167, lacking references in lines 184-187 and 209-211, and an extra right parenthesis found in line 179. An input error from "Figure3E" to "Figure 3J" in line 195 and a mislabeling of the scale bar in Figure 2C were also observed. The authors need to carefully review the entire text.

5. The reference section requires improvement, including consistent capitalization of literature titles and the number of authors throughout.

REVIEWER COMMENTS

Reviewer #1 (Remarks to the Author):

Zhang et al reported a unique X-ray triggered crosslinking polymerization. The mechanism is utilizing X-ray to activate a visible persistent luminescent and transfer emission energy to radical photoinitiator, which will generate radicals for the further crosslinking polymerization of di-vinyl monomers. Due to deep penetration of X-ray, the authors demonstrated this crosslinking polymerization can happen in deep tissue and bone tissue, which is far superior than other UV/vis or NIR photopolymerization methods. This is a very interesting study. It will obviously attract the interest from polymer scientists and biomedical engineers. It will bring significant value to the research community. Also, the synthesis of the composite and in-vitro and in-vivo experiments for hydrogel formation using photo-crosslinking polymerization are well designed and sufficient to support the conclusion. The only question and concern from the current reviewer is the polymerization efficiency.

Response: Thanks for your valuable comments. We have performed additional experiments to investigate the polymerization efficiency. Please see the detailed point-to-point responses in the following.

Major concern:

1. Given this is a process of radical polymerization and the radical is generated by X-ray activation, the efficiency of the X-ray activated polymerization should be measured by the polymerization rate of the monomers, or radical generation rate. The dosing of the composite (>5%, w/v) and CQ photoinitiator (3%) are relatively high compared to most visible light triggered photocatalytic polymerization systems (<0.1% or even ppm photocatalysts). The polymerization rate can be evaluated by monomer conversion versus irradiation time, in which monomer conversion can be easily measured by ¹HNMR. However, the monomer should be a mono-vinyl olefin (such as PEG acrylate) instead of di-vinyl. In the current study, the authors used di-vinyl PEGDA to make hydrogel which will be difficult to measure the monomer conversion. Even <5% monomer was converted, the solution at 40% PEGDA will still gel up. The low G' and G'' (< 1kPa) for all the hydrogels may suggest very low monomer conversions. Therefore, the authors should give a sense how efficient the radical will generate (or how fast the polymerization will be) from this X-ray activated process.

Response: Thank you for your valuable suggestion regarding the measurement of the efficiency of X-ray activated polymerization. In response, we have conducted further NMR analysis to evaluate monomer conversion at various irradiation intervals. To address your concern about using a di-vinyl monomer, we replaced PEGDA with a mono-vinyl olefin, poly(ethylene glycol) methyl ether acrylate (PEGMA).

The experimental procedure was similar to that in Sec. 3.2, with adjustments in irradiation times. Following irradiation, the samples were stored in a dark place at room temperature for 24 hours before NMR analysis. In the ^1H NMR spectra, (Figure S14 and S15), we specifically analyzed the signals from the protons in the vinyl groups (H_{a1} , H_{a2} , and H_b) and the methylene group (H_c) adjacent to the ester bond. These signals do not overlap with those from triethanolamine or MCD/CQ, allowing for accurate calculation of the monomer conversion.

The conversion was calculated using the formula:

$$\text{Conversion}(\%) = \left[1 - \frac{(I_{a1} + I_{a2} + I_b)/3}{I_c/2} \right] \times 100\%$$

(Equation S1)

where I_{a1} , I_{a2} , I_b , and I_c represent the integral values from H_{a1} , H_{a2} , H_b , and H_c respectively. Our results showed monomer conversion rates of 5.3%, 45.0%, 46.0%, and 49.0% for irradiation times of 1 min, 3 min, 5 min, and 10 min, respectively, indicating an increase in monomer conversion with longer irradiation times. Notably, a significant increase in conversion was observed between 1 min and 3 min, after which the conversion rate increased more gradually.

This analysis confirms the efficiency of the radical generation in the X-ray activated polymerization process and provides a clearer understanding of the polymerization kinetics under varying irradiation times.

Figure S14. ^1H NMR spectra of PEGMA, triethanolamine, and MCD/CQ in D_2O .

Figure S15. ^1H NMR spectra of the D_2O solution containing $\text{HNTs@YF}_3\text{:Tb}^{3+}$, PEGMA, triethanolamine, and MCD/CQ with exposure to X-ray for different times.

The above-mentioned results and discussion have been added in Sec 2.3 in the revised manuscript and Sec. S2.1 in the revised supplementary information.

2. The control experiments in Figure 3A to demonstrate necessity of all four components need more convincing data from ^1H NMR or others. The GPC (Figure S9) is known not sensitive to low amount of impurity in polymers. This data is important because the authors should rule out other pathways for radical generation, such as direct radical generation from X-ray/CQ and bypassing the composite.

Response: We appreciate your insightful suggestion regarding the need for more convincing data to support the control experiments in Figure 3A. In response, we have conducted additional experiments and analyzed them using $^1\text{HNMR}$ in regard to the polymerization behavior with a mono-vinyl olefin, poly(ethylene glycol) methyl ether acrylate (PEGMA).

To rigorously test the polymerization process, we set up the following experimental groups: v: PEGMA alone; vi: PEGMA with MCD/CQ and triethanolamine; vii: PEGMA with HNTs@YF₃:Tb³⁺; viii: PEGMA with MCD/CQ and HNTs@YF₃:Tb³⁺; ix: PEGMA with MCD/CQ, triethanolamine, and HNTs@YF₃:Tb³⁺. The conditions for these experiments were consistent with those described in Sec. 3.2. The $^1\text{HNMR}$ results, shown in Figure S16, indicate no significant monomer conversion (approximately 0%) in groups v, vi, vii, and viii. This finding confirms that polymerization does not occur in the absence of any of the necessary components and that radicals are not generated through X-ray/CQ interaction or by bypassing the composite. These results are consistent with our findings from GPC curves. Significantly, only in group ix, which included all required components along with X-ray activation of HNTs@YF₃:Tb³⁺, did we observe successful free radical polymerization. This comprehensive set of experiments and the corresponding NMR data effectively rule out other pathways for radical generation and strongly support the necessity of all four components for the polymerization process.

Figure S16. ^1H NMR spectra of the samples in parallel experiments. A, Group v: PEGMA. **B,** Group vi: PEGMA + MCD/CQ + triethanolamine. **C,** Group vii: PEGMA + HNTs@YF₃:Tb³⁺. **D,** Group viii: PEGMA + MCD/CQ+ HNTs@YF₃:Tb³⁺.

The above-mentioned results and discussion have been added in Sec 2.3 in the revised manuscript and Sec. S2.1 in the revised supplementary information.

Minor comments:

3. Figure 2c: the scale bar is not clear enough, which shows 100 nm in the image but was claimed as 200nm.

Response: We apologize for the confusion caused by the incorrect scale bar annotation in Figure 2c. We have corrected this mistake, and the scale bar now accurately reflects 100 nm, in line with the TEM images. For the reviewer's convenience, we have included the revised Figure 2C with the correct scale bar annotation in this submission.

Part of Figure 2. Structural and optical characterization of HNTs@YF₃:Tb³⁺. A&B, TEM images of pristine HNTs. C, TEM image HNTs@YF₃:Tb³⁺. D, HETEM image of HNTs@YF₃:Tb³⁺.

4. Figure 2F: ‘HNTs@NaYF₄:Tb³⁺’ or ‘HNTs@YF₃:Tb³⁺’, is it a typo? It also appears in supporting information four times.

Response: Such mistake has been corrected in the revised manuscript and supplementary information. It should be ‘HNTs@YF₃:Tb³⁺’, not ‘HNTs@NaYF₄:Tb³⁺’.

5. ‘m/v’ and ‘W/v’ are both used in the paper, which should be maintained consistent.

Response: We use “m/v” for all case in the revised manuscript.

6. The composite materials were used differently in in-vitro (5% suspension) and in-vivo (10%) cases. What is the concern to use different amounts?

Response: The decision to use different concentrations of composite materials in in-vitro (5% suspension) and in-vivo (10%) experiments was based on the consideration of biological dilution in the in-vivo environment.

7. Page 14, line 409, Experimental section 3.2: ‘10 min’ is the total irradiation duration or total gelation processing time?

Response: Thank you for pointing out this ambiguity. In Experimental section 3.2, the ‘10 min’ mentioned refers to the total irradiation duration. We have made this clearer in the revised version of the manuscript.

8. What is the standard in Figure S9? Are there any peaks at high *M_w* area (0-6 min elution time)? Full spectra for all three samples should be provided.

Response: Thank you for your query regarding the standards in Figure S9 and the peak details. We have now included the full spectra for all three samples in the revised supplementary information. As per your request, we confirm that there are no peaks in the high molecular weight (*M_w*) area (0-6 min elution time). Additionally,

we have added details about the standard used in the analysis, which is polystyrene with a molecular weight (M_p) of 2755 g/mol. This updated figure is now labeled as Figure S11B in the revised supplementary information. For ease of review, we are also enclosing the updated Gel Permeation Chromatography (GPC) curves here.

Figure S11B. GPC elution curves of samples i, ii, and iii in parallel experiments

Reviewer #2 (Remarks to the Author):

The manuscript still needs improvement.

The study by Zhang et al. presented a novel approach to hydrogel formation using low-dose X-ray-activated persistent luminescent phosphor through in situ photocrosslinking activation. The proposed HNTs@YF₃:Tb³⁺ nanocomposites demonstrated excellent water dispersibility and X-ray-excited luminescence. Hydrogel-forming capability of the HNTs@YF₃:Tb³⁺ upon X-ray was confirmed through in vitro and in vivo experiments. However, the manuscript lacks adequate explanations and validations for key scientific inquiries, and the presented data are insufficient to substantiate the proposed mechanisms and relevant conclusions in the research. Therefore, a major revision of the manuscript is necessary.

Response: We appreciate your feedback and acknowledge the need for further clarification and validation in our manuscript. To address these concerns, we have conducted additional experiments and provided more comprehensive data to support the proposed mechanisms and conclusions of our study. Detailed responses to each point raised are included in the following sections.

1. The introduction did not sufficiently highlight the innovation and superiority of this work. While previous researchers have reported in situ hydrogel formation using X-ray irradiation-induced polymerization (e.g., *Soft Matter*, 2012, 8(5): 1420-1427), the concept of "Xcrosslinking" proposed in this study was meaningful. The authors are encouraged to clarify the innovation and superiority of their work.

Response: Thank you for your insightful comment. We recognize the importance of highlighting the innovation and superiority of our "Xcrosslinking" approach in the introduction. To address this, we have now included a more detailed comparison with previous methodologies, specifically the one reported in *Soft Matter*, 2012, 8(5): 1420-1427. In the cited literature, X-ray irradiation directly triggers polymerization through high-energy homolysis reactions, requiring a significant X-ray dose (3.3×10^3 to 1.98×10^4 Gy). Conversely, our "Xcrosslinking" method involves a cascade reaction where X-ray activates the luminescence and afterglow of X-PLNPs, which then initiates photoinitiator-based radical generation for polymerization. This approach not only requires a substantially lower X-ray dose (only 1.5 mGy, a million times less than the previous method) but also offers enhanced controllability and the potential to integrate with other photochemical reactions.

We have elaborated on this comparison and the advantages of "Xcrosslinking" in the caption of Scheme S1 in the supplementary information.

Scheme S1. Different approaches to generate free radical polymerization via X-ray irradiation. i, In this process, the free radicals are mainly generated by the X-ray-induced homolysis reactions in which high energy is essentially needed. For example, the total dose used in the literature by using this approach is calculated as 3.3×10^3 Gy to 1.98×10^4 Gy (*Soft Matt* **8**, 1420-1427 (2012)). **ii,** The "Xcrosslinking" proposed in this study can be classified into a kind of cascade reaction, in which the X-ray is used to activate the luminescence and afterglow from X-PLNPs. The luminescence and afterglow can be further used to activate the photoinitiators and then generate free radicals for polymerization. X-PLNPs can be activated in a much lower X-ray dose than that used in inducing homolysis reactions.

2. Safety evaluation upon low-dose X-ray exposure with 30 mA and 40 kV (in line 219) and a comparison with typical clinical X-ray parameters (in line 286) should be addressed. Furthermore, selection of HeLa cells for assessing HNTs@YF₃:Tb³⁺ cytotoxicity should be justified. Additionally, toxicity evaluation of HNTs@YF₃:Tb³⁺ upon X-ray exposure and its impact on normal cells need to be discussed.

Response: Thank you for the insightful suggestions. We agree that a thorough safety evaluation is crucial. In our study, the X-ray dose (30 mA, 40 kV) used was 0.09 Gy/s. This intensity is indeed higher than typically employed in in vivo studies, but was chosen to specifically isolate the effects of the "Xcrosslinking" systems. The intent was to eliminate confounding factors such as direct radical generation from X-ray or X-ray/CQ interactions, and to focus on the role of luminescence in the process.

Regarding the selection of HeLa cells for cytotoxicity assessment, we chose them due to their widespread use in initial cytotoxicity screenings, providing a baseline for comparison with other studies. However, we acknowledge the importance of evaluating the impact on normal cells. To this end, we have used mouse fibroblast

L929 cells to assess the toxicity of HNTs@YF₃Tb³⁺ and its interaction with X-ray. We evaluated cell viability over a range of concentrations (10 to 1000 μg mL⁻¹) over 7 days. The results showed that the HNTs@YF₃Tb³⁺ exhibit negligible biological toxicity, maintaining over 90% cell viability even at the highest concentration tested.

Furthermore, when these L929 cells were exposed to X-ray (1.5 mGy) in the presence of HNTs@YF₃Tb³⁺, they also demonstrated high cell viability (> 90%). In addition to X-ray exposure, we examined the impact of green light emitted by LED, which has a higher intensity than the luminescence generated by HNTs@YF₃Tb³⁺ under X-ray conditions. The cell viability remained above 90% in these conditions as well, underscoring the negligible biological toxicity of the HNTs@YF₃Tb³⁺, the X-ray exposure, and the emitted green light on mouse fibroblast L929 cells.

We believe these results provide an enhanced overview of the safety profile of HNTs@YF₃Tb³⁺ in both cancerous and normal cell lines and under various exposure conditions.

Figure 4A. Cell viability data in mouse fibroblast L929 cells.

The results mentioned above and the discussion have been added in Figure 4A and Sec 2.4 in the revised manuscript. Detailed methods have been added in Sec. 3.3.2 in the revised manuscript. To simplify the main text, the results of Hela cells have been moved to the revised supplementary information.

3. The manuscript lacked in-depth discussions. For instance, NMR characterization towards MCD & MCD/CQ in section 2.2 and experimental data to confirm gelation formation in sections 2.3 and 2.5 were insufficiently described. Furthermore, rationale behind choosing triethanolamine as the hydrogen donor and mechanism of the free radical and hydrogel formations with X-ray radiation require detailed explanations and relevant experimental validations.

Response: Thank you for your valuable feedback. We recognize the importance of detailed discussions and experimental validations in our manuscript. We chose triethanolamine as the hydrogen donor based on its established use as a co-initiator in photopolymerizations, noted for its low oxidation potential and energy barrier (Polym Chem 6, 6595-6615 (2015)). This, combined with camphorquinone (CQ), has shown effective results in previous studies for photocrosslinked hydrogels.

For NMR analysis of MCD & MCD/CQ, our data align with previously reported studies (Polym Adv Technol 20, 723–728 (2009)), confirming successful preparation. We expanded our NMR analysis to include monomer conversion rates at different irradiation intervals, using poly(ethylene glycol) methyl ether acrylate (PEGMA) as a substitute for PEGDA. This detailed analysis, including a comparison of ¹H NMR spectra, is presented in Figures S14 and S15(please see our response to Reviewer #1). Additional experiments were also conducted to elucidate the "Xcrosslinking" system's mechanism. These experiments involved varying components (PEGMA, MCD/CQ, triethanolamine, and HNTs@YF₃:Tb³⁺) and irradiation times to assess monomer conversion and the necessity of each component in the polymerization process. The findings, detailed in Figure S16, demonstrate that successful free radical polymerization requires the combination of all components and X-ray activation.

Figure S16. ^1H NMR spectra of the samples in parallel experiments. A, Group v: PEGMA. B, Group vi: PEGMA + MCD/CQ + triethanolamine. C, Group vii: PEGMA + HNTs@YF₃:Tb³⁺. D, Group viii: PEGMA + MCD/CQ+ HNTs@YF₃:Tb³⁺.

To confirm hydrogel formation, we examined the swelling behavior of the synthesized hydrogels. After thorough dialysis and drying, we measured the swelling degree over time, calculating the equilibrium swelling degree (ESD) and applying the Fickian diffusion model to understand the swelling kinetics. The swelling behavior, as shown in Figure S13, aligns with a Fickian diffusion mechanism, with rapid initial swelling followed by a plateau, supporting the successful formation of hydrogels. In particular, we have evaluated the swelling behavior of the synthesized hydrogels by referring to the literature (*Adv. Funct. Mater.* **29**, 1904886 (2019)). Briefly, an accurately weighed dry hydrogel piece (W_0) was placed into deionized water at room temperature. The swollen samples were picked out from the medium at specific time intervals, wiped, weighed (W_i), and then placed back into the medium. The

measurements were carried out in three parallel groups to calculate the mean value and standard deviation.

The swelling degree (SD_t) at various time intervals was calculated as:

$$SD_t (\%) = \frac{W_i - W_0}{W_0} \times 100\%$$

(S2)

where i represents the swollen time, W_i represents the weight of the swollen sample at different times and W_0 is the original weight of the dry hydrogel piece. The swelling behaviors were monitored until the W_i reached a constant value which is defined as the equilibrium swelling degree (ESD).

The Fickian diffusion model was employed to gain insight into the swelling kinetics.

$$\frac{M_t}{M_\infty} = \frac{SD_t}{ESD} = k \times t^n \quad (S3)$$

$$\log \frac{SD_t}{ESD} = n \log t + \log k \quad (S4)$$

where t and k represent time and a constant, respectively; M_t and M_∞ are the amount of solvent absorbed at time t (h) and at equilibrium, respectively; n is the swelling exponent, also known as the swelling exponent, is calculated based on the slope of $\log (SD_t/ESD)$ vs. $\log t$ ($0 < SD_t/ESD \leq 0.6$).

The swelling behavior of the hydrogel prepared in Sec. 2.3 was carefully investigated by plotting SD_t vs time. SD_t increases rapidly at the first 50 min and then reaches a plateau after 120 min (case of “600 s” shown in Figure S13). The ESD is calculated as 3.29. The swelling exponent (n) is calculated as 0.400, suggesting a Fickian diffusion mechanism (*Polymer*, **45**, 6431-6435 (2004)).

Figure S13. Swelling studies. **A**, SD –time plots of the obtained hydrogel after exposure to X-ray with different times. **B**, The illustration of ESD results.

The above-mentioned results and discussion have been added in Sec 2.3 in the revised manuscript and Sec. S2.1 & S2.3 in the revised supplementary information.

Moreover, the discussion in Sec. 2.2 and 2.5 has also been enriched. We believe these additional experiments and analyses address the concerns raised and provide a more comprehensive understanding of the mechanisms at play in our "Xcrosslinking" system.

4. Numerous formatting and mistake issues were found in the manuscript, such as figure caption format in Scheme Figure & Figure 5F, missing space between value and unit in Figure 1G, lacking "Figure S5-S7" in appropriate positions within lines 161~167, lacking references in lines 184-187 and 209-211, and an extra right parenthesis found in line 179. An input error from "Figure3E" to "Figure 3J" in line 195 and a mislabeling of the scale bar in Figure 2C were also observed. The authors need to carefully review the entire text.

Response: We appreciate your attention to detail in identifying the formatting and typographical errors in our manuscript. We have addressed each issue as follows: 1. The caption formats for Scheme Figure and Figure 5F have been corrected to align with the journal's guidelines. 2. The missing space between value and unit in Figure 1G has been added for clarity and consistency. 3. We have included the references to "Figure S5-S7" in the text, which are now labeled as Figure S6-S8, in the appropriate sections (lines 161-167). 4. The necessary references have been added in the manuscript: Ref. 42 in Section 2.2 and Ref. 43 in Section 2.3. 5. The input error from "Figure 3E" to "Figure 3J" in line 195 and the mislabeling of the scale bar in Figure 2C have been corrected. In addition to these specific corrections, we have conducted a thorough review and polishing of the entire text to ensure clarity and accuracy throughout the manuscript. We hope that these revisions meet the reviewer's expectations and standards.

5. The reference section requires improvement, including consistent capitalization of literature titles and the number of authors throughout.

Response: Thank you for this comment. We have carefully checked the requirement of the "Brief guide for submission to Nature Communications". Additionally, we also have downloaded the literature recently published in Nature Communications. The reference section in the newly revised manuscript is re-written according to the following principles based on the "Brief guide" and the recently published articles:

i) These only contain citations and should only be one publication with each number. Include the title of the cited article or dataset ("Brief guide for submission to Nature Communications").

ii) The titles cited in the reference section are written in a "sentence-style capitalization" model, in which only the initial letter is in the upper case. Some words for special uses are also written in the upper case, such as "3D", "X-ray", "TADF", etc.

iii) For author lists, if 6 or more authors, only the first author is listed and

abbreviated with “*et al.*”. All author names are listed if the number of authors is equal to or less than 5.

iv) Given names are abbreviated in the reference list. The names are separated by comma symbols. The journal names are written in abbreviated and italics following the style of “*Nat Commun*”.

Reviewer #3 (Remarks to the Author):

To address the limitations of current photo-crosslinking methods in achieving deep-tissue penetration, this study designed a X-ray-controlled polymer cross-linking method utilizing HNTs@YF₃:Tb³⁺ as a visible luminescent phosphor. This study to some extent expanded the use of hydrogels in biomedical applications. Although the findings are interesting, there are some shortcomings in presenting the significance and novelty of this work. Some data should be further analyzed and even more data should be supplemented to improve this manuscript. The following issues need to be clarified.

Response: Thanks for your valuable comments. We have performed additional experiments and discussions to improve this manuscript. Please see the detailed point-to-point responses in the following.

1. The author only presented the gelatinization mechanism and environment of the hydrogel, without delving into specific application scenarios. Consequently, the scope of its applicability remains unclear, hindering a comprehensive evaluation of its clinical significance. It is suggested to supplement specific application scenarios.

Response: Thank you for your constructive suggestion. In response, we have expanded our discussion to include specific application scenarios for the “Xcrosslinking” system. A detailed illustration has been added to provide a clearer understanding of its potential uses. This is represented in the newly added Figure 6 in our manuscript.

We envision some possible applications for this system:

1. **Deep Tissue Repairs:** The system offers prospects for noninvasive 2D or 3D printing, aiming to repair damaged deep tissues in situ, potentially eliminating the need for surgical intervention.
2. **Dental Repairs:** The X-PLNPs can be activated by X-ray, utilizing the resultant luminescence as a “Nanolight” source to initiate curing processes in photosensitive resins, thus reducing patient discomfort during prolonged dental procedures.
3. **Embolotherapy for Tumors:** We anticipate the use of the “Xcrosslinking” system in endovascular embolotherapy for malignant tumors. By precise injection and X-ray exposure, gelatinization within tumor vessels can be induced, potentially leading to an effective embolotherapy outcome.
4. **Cartilage Defect Repairs:** The system could also be adapted for cartilage defect repair, with the crosslinking degree and swelling behavior being adjustable through irradiation time and X-ray doses, tailoring it to the specific mechanical requirements of cartilage applications.

These application scenarios have been included in the Conclusion section of our revised manuscript, and the detailed illustration is now presented as Figure 6. We believe these additions will significantly enhance the clarity of our manuscript and

provide a better understanding of the potential clinical significance of the “Xcrosslinking” system.

Figure 6. Summary of the potential application scenarios of the “Xcrosslinking” system. A, Deep tissue repairs. B, Dental repairs. C, Embolotherapy on tumors. D, Cartilage defect repairs.

2. Currently, there exist several mechanisms to confer injectability upon hydrogels and facilitate their gelation within deep tissues. Examples include dynamic covalent hydrogels, temperature-sensitive hydrogels, shear-thinning hydrogels, and more. These hydrogels can undergo gelation without relying on external devices to initiate cross-linking after injection. In comparison to the aforementioned hydrogels, what are the distinct advantages of X-ray-activated hydrogels in medical applications? The authors should provide a more specific discussion to elucidate the innovative aspects of this study.

Response: Thank you for your insightful suggestion. We have incorporated a detailed comparison between X-ray-activated hydrogels and other injectable hydrogels into the Introduction of our revised manuscript. Additionally, here is a summary for your reference:

“While various mechanisms exist for making hydrogels injectable and capable of in situ gelation, such as temperature sensitivity, pH responsiveness, magnetism, shear-

thinning, and chemical triggers, each has its limitations. For instance, temperature-sensitive hydrogels may gel prematurely in the syringe due to body heat (Biomacromolecules 6, 2930-2934 (2005)), magnetic stimuli might interfere with certain clinical treatments and imaging techniques (Sci Adv 6, eaay0065 (2020)), and pH-responsive hydrogels are limited to environments matching their pH sensitivity (Nat Mater 14, 1065–1071 (2015)). Chemical stimuli lack spatial control, and while shear-thinning hydrogels are injectable, controlling gelation post-injection is challenging (Adv Mater 32, 1906012 (2020)). In contrast, our “Xcrosslinking” strategy, utilizing X-ray activation, overcomes many of these challenges. It offers precise spatial control of gelation, operates effectively as a non-contact external stimulus, and is less dependent on the surrounding environmental conditions. This approach allows for on-demand gelation within deep tissues, which is a significant advantage in medical applications where precise localization and minimally invasive procedures are crucial. These distinctions underscore the innovative aspects of the X-ray-activated hydrogels, positioning them as a superior choice for certain medical applications where traditional stimuli-responsive hydrogels may fall short.”

We believe this addition provides a clear differentiation of our X-ray-activated hydrogels, highlighting their unique advantages in the realm of medical applications.

3. The statement “Our results demonstrated high biocompatibility and the potential safety of this approach, as it enabled depth-independent in situ crosslinking activation.” in the introduction raises the question of whether the cross-linking system employed in this study is truly unaffected by tissue depth. Substantiating this conclusion necessitates additional experimental evidence.

Response: Thank you for highlighting this important aspect. We have further investigated the influence of the tissue depth on the radioluminescence behaviors of the prepared HNTs@YF₃:Tb³⁺. After exposure to X-ray (40 kV, 30 mA) for 10 min, the peak intensity at 545 nm can reach a constant value and was recorded as I_0 . Then the tissue samples with a certain thickness were placed between the X-ray source and HNTs@YF₃:Tb³⁺ sample. The peak intensity at 545 nm was recorded as I_t after it reached the plateau. The intensity ratio (I_t / I_0) was calculated and the relationship between intensity ratio and tissue thickness is shown in Figure S20. Two different kinds of tissues from chicken breast and bovine bones were used in this study.

Figure S20. Relationship between intensity ratio (I_t / I_0) and tissue thickness. A, Plots of intensity ratio vs. thickness in different tissues. **B,** Linear relationship between log [intensity ratio] and thickness. Two different kinds of tissues from chicken breast and bovine bones were used in this study. Chicken breast pieces from 4 to 40 mm and bone tissue from 4 to 20 mm were regarded as soft tissue and bone tissue, respectively.

Our results showed radioluminescence can be detected even with 4 cm soft tissue and 2 cm dense bovine bone but a graduate decrease in intensity ratio with increasing tissue thickness. The data is presented in Figure S20, which includes plots of intensity ratio versus thickness and a linear relationship between log [intensity ratio] and tissue thickness. Based on these findings, we acknowledge that our initial claim of "depth-independent" crosslinking was overly broad. In the revised manuscript, we have changed this to "in situ crosslinking activation in deep tissue." While the penetrability of our system is substantial and surpasses that of UV, visible light, and near-infrared light-based systems, we recognize it is not entirely unaffected by depth. This discussion and the new experimental result have been included in Section S2.5 of the supplementary information.

4. Please provide further details regarding the rationale behind selecting HNTs as the substrate in this study and elucidate their unique advantages over other materials. Additionally, please illustrate the decision to employ a hollow structure rather than a solid nanorod, and discuss the implications of this choice on the overall impact of the study.

Response: Thank you for this comment. We have added more description of HNTs in the introduction part to further elucidate their unique advantages.

"HNTs are natural tubular nanomaterials with good water dispersibility. The hollow tubular structure of HNTs effectively reduces the density. The stable negative charge on the external surface effectively prevents the aggregation of the nanotubes. As a result, HNTs exhibit significantly better water dispersibility than those of commonly used nanoparticles. Other attractive properties including good biocompatibility, low

toxicity, non-degradation, hydrophilic, processability, and low-cost also make HNTs promising materials in the biomedical field. Moreover, the existence of -OH groups on the surface makes them accessible to be functionalized.”

The above-mentioned details have been added in the Introduction part in the revised manuscript.

To demonstrate the good water dispersibility of HNTs, we made a comparison of HNTs and commercially available attapulgite nanorods (ATPs). ATPs are natural rodlike nanomaterials with a similar size to HNTs. As shown in the Figure S1, HNTs display good dispersibility in water. The emitted bluish opalescence demonstrates this point. However, obvious precipitations can be found in the ATPs-containing water solution. The upper and lower parts show different transparency, suggesting a limited water-dispersibility of ATPs. The unsatisfied water-dispersibility would give rise to uniform hydrogels and thereby result in limited mechanical properties. The better water-dispersibility of HNTs than solid nanorods makes HNTs a superior candidate for fabricating hybrid hydrogels.

Figure S1. Photographs of the water solution containing different nanomaterials (concentration: ca. 1.0 mg/mL). A, Halloysite nanotubes (HNTs). B, Commercially available attapulgite nanorods (ATPs).

5. The authors should explore the impact of PEGDA's molecular weight and content on both gelation time and rheological properties of the hydrogel system. Furthermore, it is essential to determine whether PEGDA with a molecular weight of 400 represents the optimal component.

Response: This is an excellent suggestion. We have explored the impact of PEGDA's molecular weight (200, 400, and 600 g/mol) and content on gelation time and rheological properties. Here, we use PEGDA₂₀₀, PEGDA₄₀₀, and PEGDA₆₀₀ to represent PEGDA with molecular weights of 200, 400, and 600 g/mol, respectively. The reaction conditions are the same as those used in Sec. 3.2 except for changing the irradiation times. PEGDA₂₀₀, PEGDA₄₀₀, and PEGDA₆₀₀ can be gelled within 60 s, 180 s, and 360 s, respectively, suggesting the gelation time increases with the increase

of the PEGDA's molecular weight. However, it should be noted that PEGDA₂₀₀ cannot be dissolved in water, which yields a nonuniform hydrogel (Figure S12A) and is not considered for further studies. PEGDA₄₀₀ and PEGDA₆₀₀ can be dissolved in water at any proportion. The obtained PEGDA₄₀₀ and PEGDA₆₀₀ hydrogel samples (Figure S12B&C) were cut into desired sizes and then subjected to rheological analysis. The rheological characteristics of the resulting hydrogel were demonstrated by the storage modulus (G') being greater than the loss modulus (G'') over an average frequency range of 1 to 100 rad/s (Figure S12D). The PEGDA₆₀₀ hydrogel shows higher G' and G'' than the PEGDA₄₀₀ hydrogel.

Figure S12. Photographs and rheological results of the hydrogels by using PEGDA with different molecular weights. A-C, Photographs of the hydrogel samples by using PEGDA with different molecular weights (200, 400, and 600 g/mol). D, Rheological property of the as-formed hydrogels.

The main purpose of this study is to develop an “Xcrosslinking” approach for potential *in vivo* use. We regarded PEGDA₄₀₀ as the optimal component because it can be dissolved in water and is able to achieve a gelation state in a shorter time than that of a higher molecular weight. Furthermore, the PEGDA₄₀₀ content (20, 30, and 40%) on gelation time was examined. HNTs@YF₃:Tb³⁺ cannot achieve good dispersity when the PEGDA₄₀₀ content is higher than 50%. The results indicate that the gelation can be achieved within 600 s, 360 s, and 180 s as the PEGDA₄₀₀ content ranges from 20, 30, to 40%, respectively, implying the gelation time decreases as the increase of PEGDA₄₀₀ content and the content of 40% was used for further investigations.

The above-mentioned results and discussion have been added in Sec. S2.2 in the revised supplementary information.

6. The author should supplement additional experiments to elucidate the swelling behavior of the hydrogel following irradiation-induced gelatinization, as the utilization of these hydrogels in vivo entails exposure to a significant volume of bodily fluids.

Response: Thank you for the comment. The swelling behavior of the hydrogel has been investigated. Details are shown in the following:

The obtained hydrogel was dialyzed against deionized water thoroughly to remove the unreacted residues and then dried at room temperature. We have evaluated the swelling behavior of the synthesized hydrogels by referring to the literature (*Adv. Funct. Mater.* **29**, 1904886 (2019)). Briefly, an accurately weighed dry hydrogel piece (W_0) was placed into deionized water at room temperature. The swollen samples were picked out from the medium at specific time intervals, wiped, weighed (W_i), and then placed back into the medium. The measurements were carried out in three parallel groups to calculate the mean value and standard deviation.

The swelling degree (SD_t) at various time intervals was calculated as:

$$SD_t (\%) = \frac{W_i - W_0}{W_0} \times 100\% \quad (S2)$$

where i represents the swollen time, W_i represents the weight of the swollen sample at different times and W_0 is the original weight of the dry hydrogel piece. The swelling behaviors were monitored until the W_i reached a constant value which is defined as the equilibrium swelling degree (ESD).

The Fickian diffusion model was employed to gain insight into the swelling kinetics.

$$\frac{M_t}{M_\infty} = \frac{SD_t}{ESD} = k \times t^n \quad (S3)$$

$$\log \frac{SD_t}{ESD} = n \log t + \log k \quad (S4)$$

where t and k represent time and a constant, respectively; M_t and M_∞ are the amount of solvent absorbed at time t (h) and at equilibrium, respectively; n is the swelling exponent, also known as the swelling exponent, is calculated based on the slope of $\log (SD_t/ESD)$ vs. $\log t$ ($0 < SD_t/ESD \leq 0.6$).

The swelling behavior of the hydrogel prepared in Sec. 2.3 was carefully investigated by plotting SD_t vs time. SD_t increases rapidly at the first 50 min and then reaches a plateau after 120 min. The ESD is calculated as 3.29. The swelling exponent (n) is calculated as 0.400, suggesting a Fickian diffusion mechanism (*Polymer*, **45**, 6431-6435 (2004)). For review's convenience, we have added the swelling curve in the following (case of "600 s" in Figure S13).

The above-mentioned details have been added in Sec. S2.3 in the revised supplementary information.

Additionally, we have investigated the effect of radiation time on swelling behavior. Detailed please see the response to Comment 9.

Figure S13. Swelling studies. **A**, SD–time plots of the obtained hydrogel after exposure to X-ray with different times. **B**, The illustration of ESD results.

7. Please confirm whether there are unreacted monomers following the gelation of the reaction stock solution *in vivo*. Will the hydrogel degrade *in vivo*? Moreover, are there any potential adverse effects associated with long-term retention of the hydrogel in the body, particularly within the bones?

Response: We have performed additional experiments to calculate the conversion of the hydrogels, which is measured as ca. 80~90% in *in-vitro* studies. Therefore, there should be some unreacted monomers following the gelation of the reaction stock solution *in vivo*. It should be noted that polyethylene glycol diacrylate (PEGDA) and PEGDA hydrogels have been approved by the FDA for several biomedical applications owing to their low toxicity and high biocompatibility (*Adv Mater* **32**, 2001459 (2020)). Because of the presence of ester bonds, the three-dimensional network is able to degrade *in vivo* (*Biomacromolecules* **13**, 779–786 (2012); *Polym Chem* **11**, 568-580 (2020)). Numerous studies focusing on the degradation of PEGDA *in vitro* and *in vivo* have been conducted in recent years. Up to now, no serious adverse effects have been reported. In this paper, we focus on developing an “Xcrosslinking” approach to fabricate gelation within deep tissues. Besides PEGDA, other di- or multi-vinyl olefins can also be used in this system to meet the various needs of different tissues.

8. Although gelatinization within the bone is achievable, it requires an extended exposure time under X-ray radiation. However, it remains unclear whether this exposure condition is within the acceptable range of safety. The mention of “low-dose X-ray irradiation” lacks specificity and requires further clarification. Please provide the safe dose for X-ray irradiation and present the corresponding results and discussions on the gelation ability of the system in bone under the safe irradiation dose.

Response: The total X-ray dose used in gelatinization within the bone is calculated as 110 Gy. It has been reported that X-ray irradiation with a dose of 110 Gy can be used in rectal cancer patients with little toxicity (*Int J Radiat Oncol* **72**, 665-670 (2008); *Clin*

Transl Radiat Oncol **24**, 92-98 (2020)). This point has been added in Sec 2.5 in the revised manuscript.

Higher X-ray doses can also be found in other studies for treating cancers (*Int J Radiation Oncol Biol Phys* **111**, 143-151 (2021); *Develop Med Child Neuro* **49**, 577-581 (2007)). Additionally, the investigation of gelatinization within the bone in this study is a preliminary attempt to test the feasibility of the “Xcrosslinking” system inside the bone tissue. Further studies will continue to reduce the X-ray dose by optimizing the luminescence properties of X-PLNPs and using an "On-Off-On" circulation approach to leverage the persistent afterglow to suppress the potential side effects caused by irradiation.

9. The article stated “The author recorded persistent luminescence decay curves after irradiation with an X-ray irradiator for 1 minute. Remarkably, the afterglow persisted for up to 50 seconds with gradual attenuation. Prolonging the irradiation time led to a noticeable increase in afterglow intensity.” It remains unclear whether there is a linear relationship between X-ray irradiation time and the duration and intensity of afterglow. It is important to investigate whether adjusting the X-ray irradiation time can accurately control the luminescence of deep internal light sources and further modulate the degree of hydrogel crosslinking.

Response: This is a good suggestion. The dependence of X-ray irradiation time (t) on the intensity of afterglow at 600 s was investigated. The plots between $\log t$ and the intensity of afterglow on 600 s were fitted linearly, in which the correlation index (R) was 0.983. The results have been added in Sec 2.2 in the revised manuscript and Figure S9 in the revised supplementary information.

We also investigate whether adjusting the X-ray irradiation time can modulate the degree of hydrogel crosslinking.

Generally, the degree of hydrogel crosslinking holds a negative correlation with swelling behaviors (*Macromolecules* **53**, 6566-6575 (2020)). For the hydrogels made from the same monomers, a higher equilibrium swelling degree (ESD) means a lower degree of hydrogel crosslinking. So, we investigated the effect of radiation time (200 s, 400 s, and 600 s) on swelling behavior in order to reveal the effect on the degree of hydrogel crosslinking. The contents and X-ray source are the same as those used in Sec. 3.2 except for changing the irradiation times.

SD_t reaches the plateau after 120 min, 300 min, and 500 min as for the case of 600 s, 400 s, and 200 s, respectively. The ESD values are calculated as 4.34, 3.52, and 3.29 as the radiation time increases from 200 s to 600s. All of the results suggest that a longer radiation time can give rise to a lower ESD value and slower swelling behavior of the obtained hydrogels, implying the degree of hydrogel crosslinking can be well-controlled by modulating the radiation time. The above-mentioned details have been added in Sec. S2.4 in the revised supplementary information.

For review’s convenience, we have added the results in the following:

Figure S9. Relationship between X-ray irradiation time and intensity of afterglow at 600 s.

Figure S13. Swelling studies. **A**, SD–time plots of the obtained hydrogel after exposure to X-ray with different times. **B**, The illustration of ESD results.

10. The examination of the effect of HNTs@YF₃:Tb³⁺ on cell proliferation limited to a 48-hour period lacks sufficient persuasive power, necessitating an extension of the research time to a minimum of 7 days. Furthermore, in the subcutaneous gelatinization experiment, while presenting low-magnification images of rat skin tissues, it is essential to supplement high-magnification images of the tissue adjacent to the hydrogel to more clearly assess the inflammatory response caused by the material. Additionally, solely evaluating the biocompatibility of HNTs@YF₃:Tb³⁺ is not enough. As previously mentioned, other existing photo-crosslinking methods may pose DNA damage risks to

cells, thus raising concerns regarding safety. Therefore, it should be worthwhile for this study to investigate the effects of X-ray radiation and stimulated visible light on cell viability.

Response: This is a professional suggestion. We have performed additional test to evaluate the cell viability of HNTs@YF₃:Tb³⁺ with the concentration ranging from 10 to 1000 μg mL⁻¹ for 7 days by using mouse fibroblast L929 cells. Moreover, the toxicity evaluation of HNTs@YF₃:Tb³⁺ upon X-ray exposure and its impact on mouse fibroblast L929 cells have been also examined.

The obtained HNTs@YF₃:Tb³⁺ also demonstrated negligible biological toxicity upon L929 cells for 7 days, as cell viability remained above 90% even at relatively high concentrations (1000 μg/mL). The HNTs@YF₃:Tb³⁺-treated L929 cells with the exposure to X-ray (1.5 mGy) also exhibit high cell viability (> 90%). Moreover, we also evaluated the impact of green light emitted by LED on the cell viability, in which the light intensity is obviously higher than the luminescence generated by HNTs@YF₃:Tb³⁺ under X-ray. A good cell viability (> 90%) can also be achieved. The results indicate that the obtained HNTs@YF₃:Tb³⁺, 1.5 mGy X-ray, and the generated green light exhibit negligible biological toxicity on mouse fibroblast L929 cells.

Figure 4A. Cell viability data in mouse fibroblast L929 cells.

The results mentioned above and the discussion have been added in Figure 4A and Sec 2.4 in the revised manuscript. Detailed methods have been added in Sec. 3.3.2 in the revised manuscript. To simplify the main text, the results of Hela cells have been moved to the revised supplementary information.

High-magnification images of the tissue are provided in Figure S18 in the revised supplementary information. No inflammatory responses can be found in the high-magnification images, which further demonstrate the excellent biocompatibility of the hydrogels. For reviewer's convenience, we have also added the images in the following:

Figure S18. Representative photomicrograph of skin tissue sections (H&E) from negative control group, positive control group and test group.

REVIEWER COMMENTS

Reviewer #1 (Remarks to the Author):

The authors have addressed the comments properly. It is recommended to publish in the journal.

Reviewer #2 (Remarks to the Author):

The authors have revised their manuscript by supplementing some experimental results and discussion, which is now suitable for publication.

Reviewer #3 (Remarks to the Author):

The authors have supplemented relevant data in response to some of my concerns. However, there are a few discrepancies between certain issues and the actual facts, which require the author's attention.

1. Currently, visible light-cured dental resin is commonly employed for repairing teeth defects, with a typical curing time of around 30 seconds. Would the author's X-PLNPs offer a faster alternative to this? In reality, the patient's mouth opening time during the tooth defect repair process is primarily dedicated to cleaning the carious cavity and shaping the resin, rather than curing the resin. Moreover, the author's proposed "Xcrosslinking" strategy for tooth restoration necessitates additional X-ray stimulation for the activation of X-PLNPs. Consequently, the "Xcrosslinking" strategy appears to be more complex compared to existing methods and does not exhibit significant advantages. The author should thoroughly consider whether adopting this application scenario is reasonable.

2. For this study, the use of dry hydrogel for the swelling test should be avoided, and instead, solidified hydrogel should be utilized. This is necessary as the intended application does not involve dry hydrogel, therefore the experiment design must align with the actual application conditions. Furthermore, in accordance with the swelling degree calculation formula provided by the author, the ordinate axis of the swelling results (Figure S13a) should be represented as a percentage.

3. Bone tissue is known to be sensitive to radiation, and excessive exposure can lead to radiation-induced bone necrosis. In the conventional radiotherapy approach, about 30 separate fractions are typically utilized for irradiation, with a total dose of 70-80 Gy representing the limit dose for bone radiotherapy. The purpose of multifraction irradiation is to gradually increase tumor cell damage while affording normal tissue cells enough time to repair. Thus, it is difficult to imagine the extent of damage that could occur to the bone if the author were to employ a single fraction of 110 Gy. Therefore, the author must carefully explain the rationale behind their proposed strategy for gelatinization within the bone.

Reviewer #1 (Remarks to the Author):

The authors have addressed the comments properly. It is recommended to publish in the journal.

Response: Thank you for your positive feedback and for recommending our paper for publication. We appreciate your support and guidance throughout the review process.

Reviewer #2 (Remarks to the Author):

The authors have revised their manuscript by supplementing some experimental results and discussion, which is now suitable for publication.

Response: Thank you for acknowledging the revisions and additional experimental results in our manuscript. We are pleased to hear that it now meets the publication standards. We appreciate your guidance and support throughout this process.

Reviewer #3 (Remarks to the Author):

The authors have supplemented relevant data in response to some of my concerns. However, there are a few discrepancies between certain issues and the actual facts, which require the author's attention.

Response: Thank you for your valuable comments. We've added detailed explanations and conducted further swelling studies, as outlined in our point-by-point responses. We will also address the noted discrepancies for accuracy.

1. Currently, visible light-cured dental resin is commonly employed for repairing teeth defects, with a typical curing time of around 30 seconds. Would the author's X-PLNPs offer a faster alternative to this? In reality, the patient's mouth opening time during the tooth defect repair process is primarily dedicated to cleaning the carious cavity and shaping the resin, rather than curing the resin. Moreover, the author's proposed "Xcrosslinking" strategy for tooth restoration necessitates additional X-ray stimulation for the activation of X-PLNPs. Consequently, the "Xcrosslinking" strategy appears to be more complex compared to existing methods and does not exhibit significant advantages. The author should thoroughly consider whether adopting this application scenario is reasonable.

Response: Thank you for your insightful comments. We acknowledge that the standard visible light-cured dental resin, typically curing in about 30 seconds, is efficient for most dental repairs. However, our X-PLNPs are proposed as a complementary method, especially for deep dental defects where visible light curing is less effective.

In deep defects, the effectiveness of visible light curing is compromised due to reduced light penetration, leading to issues like inadequate polymerization and shrinkage (Dent. Mater. 31, 93-104 (2015)). Our "Xcrosslinking" strategy with X-PLNPs can be used to overcome this by providing uniform curing irrespective of defect depth. While this method involves additional X-ray stimulation, it is designed to enhance repair quality

in deeper defects, a challenge for current methods (J. Am. Dent. Assoc. 137, 213-223 (2006); Z. Med. Phys. 30, 194-200 (2020)).

We have elaborated on this in the revised Supplementary Information and adjusted the Conclusion accordingly. Our intention is to present "Xcrosslinking" as a supplementary technique for specific scenarios, not as a replacement for existing methods (Nat. Commun. 14, 3653 (2023)).

2. For this study, the use of dry hydrogel for the swelling test should be avoided, and instead, solidified hydrogel should be utilized. This is necessary as the intended application does not involve dry hydrogel, therefore the experiment design must align with the actual application conditions. Furthermore, in accordance with the swelling degree calculation formula provided by the author, the ordinate axis of the swelling results (Figure S13a) should be represented as a percentage.

Response: Thank you for your valuable suggestion regarding the use of solidified hydrogel samples to more accurately reflect real-world application conditions. In response, we have conducted additional experiments using solidified hydrogels prepared through the "On-Off-On" circulation process. These samples were tested for swelling behavior in phosphate buffer saline (PBS) at pH 7.4 and 37 °C, aligning with the intended application environment.

The original weight of the hydrogel sample (m_0) was accurately measured before immersion in PBS. At specific intervals, the swollen samples were removed, gently wiped, weighed (m_i), and then returned to the medium. This process was repeated, and measurements were taken in triplicate to ensure accuracy and calculate the mean and standard deviation.

The swelling degree (SD_t^*) of the solidified hydrogel sample at various time intervals was calculated as:

$$SD_t^* (\%) = (m_i - m_0) / m_0 \times 100\% \quad \text{Supplementary Equation (5)}$$

where i represents the swollen time, m_i represents the weight of the swollen sample at different times, and m_0 is the original weight of the hydrogel sample. The swelling behaviors were monitored until the m_i reached a constant value which is defined as the equilibrium swelling degree measured from the solidified hydrogel sample (ESD*).

The swelling behaviors of hydrogel samples prepared following the "On-Off-On" circulation process are depicted in Supplementary Fig. 21. The sample prepared after 10 cycles (1.5 mGy) shows an ESD* of ca. 48.2%, suggesting a low swelling behavior. (*J. Mater. Chem. B* 7, 5490-5501 (2019); *Mater. Sci. Eng. C* 127, 112208 (2021); *Adv. Sci.* 10, 2303326 (2023)) Moreover, the ESD* can be further decreased by extending the cycle times, which is beneficial to meet different requirements of bio-medical applications.

For your convenience, we have included Supplementary Fig. 21 in the revised manuscript.

Supplementary Figure 21. Swelling studies of solidified hydrogel samples. a SD*–time plots of the obtained hydrogel prepared following the “On-Off-On” circulation process with different cycles (10 cycles: 1.5 m Gy; 15 cycles: 2.3 mGy; 20 cycles: 3.0 mGy). **b** The illustration of equilibrium swelling degree measured from the solidified hydrogel sample (ESD*).

The ordinate axis of the swelling results in Figure S13a, now labeled as Supplementary Fig. 13a, is presented as percentage now. Supplementary Fig. 13b is also revised accordingly.

3. Bone tissue is known to be sensitive to radiation, and excessive exposure can lead to radiation-induced bone necrosis. In the conventional radiotherapy approach, about 30 separate fractions are typically utilized for irradiation, with a total dose of 70-80 Gy representing the limit dose for bone radiotherapy. The purpose of multifraction irradiation is to gradually increase tumor cell damage while affording normal tissue cells enough time to repair. Thus, it is difficult to imagine the extent of damage that could occur to the bone if the author were to employ a single fraction of 110 Gy. Therefore, the author must carefully explain the rationale behind their proposed strategy for gelatinization within the bone.

Response: Thank you for your professional insights regarding the X-ray dose used in our "Xcrosslinking" strategy, especially concerning bone tissue's sensitivity to radiation. We recognize the critical need to address potential side effects and ensure the safety of our method. This study serves as a proof-of-concept, and we acknowledge there is ample room for improvement.

In our research, we employed a thick, dense bovine bone model, fully polymerizing the entire cavity, to provide an illustrative example. Bovine bone, known for its higher density and stronger X-ray attenuation compared to human bones, was chosen to represent a challenging scenario. In practical medical applications, bone defects are often located in more superficial layers, and the affected area is typically smaller than our model. Therefore, the X-ray dose required in real-world applications is expected to

be lower than the 110 Gy used in this conceptual study.

A key future direction involves optimizing the luminescence properties of X-PLNPs to boost performance, thereby achieving more effective crosslinking with minimal radiation exposure.

Moreover, the potential of this method extends beyond bone applications. It is also applicable in non-medical scenarios, such as repairing deep defects in materials like concretes, alloys, and composites, where the current light based method is limited (Nature 565, 343–346 (2019)).

We have included such a discussion on these points in Section 2.5 of the main text, addressing radiation exposure concerns and emphasizing the wide-ranging applications of the "Xcrosslinking" strategy.

Other revisions

Additionally, we have performed some other revisions on format. To simplify the text, the changes relating to format issues were not marked. Here, we briefly summarize the revisions in the following:

1. Panels in figures "A, B, C..." have been revised to "a, b, c...".
2. "Supplementary Figure 1" is used to replace "Figure S1". Congeneric cases have also been revised.
3. "g/mL" has been revised to "g mL⁻¹". Similar cases have also been revised.
4. Abbreviations are defined in each figure caption.

REVIEWERS' COMMENTS

Reviewer #3 (Remarks to the Author):

The authors have addressed all my concerns. Now this article is suitable for publication.